# An ancient metalloenzyme evolves through metal preference modulation

K. M. Sendra [1] ✉, A. Barwinska-Sendra[1], E. S. Mackenzie[1], A. Baslé[1],
T. E. Kehl-Fie [2,3] ✉ & K. J. Waldron [1,4] ✉

Evolution creates functional diversity of proteins, the essential building blocks of all biological systems. However, studies of natural proteins sampled across the tree of life and evaluated in a single experimental system are lacking. Almost half of enzymes require metals, and metalloproteins tend to optimally utilize the physicochemical properties of a specific metal co-factor. Life must adapt to changes in metal bioavailability, including those during the transition from anoxic to oxic Earth or pathogens' exposure to nutritional immunity. These changes can challenge the ability of metalloenzymes to maintain activity, presumptively driving their evolution. Here we studied metal-preference evolution within the natural diversity of the iron/manganese superoxide dismutase (SodFM) family of reactive oxygen species scavengers. We identified and experimentally verified residues with conserved roles in determining metal preference that, when combined with an understanding of the protein's evolutionary history, improved prediction of metal utilization across the five SodFM subfamilies defined herein. By combining phylogenetics, biochemistry and structural biology, we demonstrate that SodFM metal utilization can be evolutionarily fine tuned by sliding along a scale between perfect manganese and iron specificities. Over the history of life, SodFM metal preference has been modulated multiple independent times within different evolutionary and ecological contexts, and can be changed within short evolutionary timeframes.

Change in the functions and properties of proteins is a fundamental process in the evolution of life[1]. So far, most functional studies of natural protein evolution have focused on changes in substrate specificity[2,3] or mechanism[3], and were mostly performed on a narrow phylogenetic sampling. Unlike changes in protein function, metal-preference evolution can reflect adaptations to changes within an existing or new niche (for example, metal availability) but with retention of the ancestral function (for example, superoxide dismutation). Many metalloenzymes are highly metal specific[4,5], especially those that use their co-factor to catalyse redox transformations, and loading of metalloproteins with suboptimal metals decreases their activity and fitness of their host organism[6–8]. Mis-metalation is a major mechanism in metal toxicity[7], neurological disorders[9] and nutritional immunity[6], and is a promising tool in combating pathogenic microbes[10]. As it is possible to experimentally evaluate environmental, cellular and protein-bound metal abundance, metal co-factor utilization provides an important and tractable system for studying natural protein evolution.

[1]Biosciences Institute, Faculty of Medical Sciences, Newcastle University, Newcastle upon Tyne, UK. [2]Department of Microbiology, University of Illinois Urbana-Champaign, Urbana, IL, USA. [3]Carl R. Woese Institute for Genomic Biology, University of Illinois Urbana-Champaign, Urbana, IL, USA. [4]Present address: Institute of Biochemistry and Biophysics, Polish Academy of Sciences, Warsaw, Poland. ✉e-mail: kacpersendra.ncl@gmail.com; kehlfie@illinois.edu; kwaldron@ibb.waw.pl

Evidence shows that metal specificities within protein families have frequently changed during their evolutionary history[11]. These changes have been important in the evolution of commensals into pathogens as they adapt to metal restriction in the host[12,13], and in the adaptation of organisms to changes in terrestrial conditions such as altered elemental bioavailability[11,14] induced by oxygenation of the atmosphere and oceans by the emergence of photosynthesis. In contrast to metal-specific proteins, metal-interchangeable (cambialistic) proteins can function using alternative metals[15,16]. Only a few metalloenzymes have been rigorously demonstrated to be cambialistic in vivo, including some iron (Fe)- or manganese (Mn)-dependent superoxide dismutases (SODs; SodFMs)[16]; metalloenzymes that play an important role in cellular reactive oxygen species defence[17].

In this Article, we sought to investigate the mechanisms and extent of metal-preference evolution across the tree of life in a widely distributed and highly conserved metalloenzyme family. To study this process, we leveraged the existence of naturally Fe-specific, Mn-specific and flexible Fe/Mn-cambialistic SodFM homologues, which differentially utilize these metals to catalyse the detoxification of superoxide.

## SodFM metal preference is not phylogenetically constrained

Bioinformatic analyses of 3,058 genomes sampled across the tree of life (146 eukaryotic, 2,613 bacterial and 281 archaeal genomes) showed that SodFMs are the most widely distributed and most highly conserved of the canonical superoxide detoxification enzymes (Extended Data Fig. 1a,b). Phylogenetics (Extended Data Fig. 2b,d), sequence (Extended Data Fig. 3a) and structural comparisons (Extended Data Fig. 2a,c,e) enabled us to subdivide the SodFM protein family into five main subfamilies: SodFM1–5 (Fig. 1a–c and Extended Data Fig. 2). Members of the two main subfamilies, SodFM1 and SodFM2, are commonly annotated in databases as being either Mn or Fe specific, respectively[14]. To test this, we heterologously expressed 64 representative SodFMs (Extended Data Fig. 4) sampled from across their protein family tree (Fig. 2a) and characterized their activity with Fe and Mn. This revealed that neither subfamily was homogeneous for metal utilization (Fig. 2a) and should no longer be referred to as Fe or Mn specific on the basis of homology searches alone. The robustness of the SodFM1 and SodFM2 groupings (Extended Data Fig. 3), and their wide distribution across the tree of life (Fig. 1a), is consistent with the hypothesis that the split between the SodFM1 and SodFM2 subfamilies is ancient[14]. On the basis of the distribution of the different metal preferences across their protein trees, ancestral sequence reconstruction (Fig. 2a) and estimations of ancestral metal utilization (Extended Data Fig. 2f,g), it seems that the ancient last common ancestors (LCAs) of SodFM1s and SodFM2s were probably Mn and Fe specific, respectively, but these ancestral states were later changed multiple independent times throughout the SodFM family's evolution (Fig. 2a).

## Uncovering untapped evolutionary diversity of SodFMs

We also identified three further subfamilies of distinct SodFMs, SodFM3–5 (Extended Data Fig. 2). SodFM3s and SodFM4s seem to be restricted mostly to prokaryotes as none were identified in our sampling of eukaryotes (Fig. 1b). SodFM4s are the most divergent of all SodFMs (Fig. 1b,c and Extended Data Fig. 2b) but are largely uncharacterized, with only a single available crystal structure (Extended Data Fig. 2a). Like the SodFM1s and SodFM2s, metal preference is not conserved across the SodFM3 and SodFM4 subfamilies (Fig. 2a). One of the key structural distinctions between these subfamilies is that SodFM1/2s are homodimeric and SodFM3/4s are homotetrameric (Extended Data Fig. 2a). The subfamily of SodFM5s provides an interesting intermediate group of tetrameric enzymes (Extended Data Fig. 2a) with some sequence features, including the frequency of $X_{D-2}$ residue (second residue towards the N-terminus from Asp metal ligand), more akin

to those of dimeric SodFM1s (Figs. 1c and 2b). SodFM5s form a sister group to SodFM3s in phylogenetic trees (Extended Data Fig. 2b,e) and structural comparisons (Extended Data Fig. 2c,d), and consist mostly of eukaryotic mitochondrial SodFMs (for example, human mitochondrial Mn-SOD) and a few bacterial sequences including that of *Parachlamydia acanthamoebae* (Fig. 2a). Grouping of the highly distinct mitochondrial SODs with sequences from *Chlamydia* could represent functional convergence, eukaryote to prokaryote lateral gene transfer (LGT) or be evidence of ancient LGT between the bacterial ancestor of mitochondria and that of *Chlamydia*.

## SodFM metal preference is continuous rather than discrete

Within subfamilies SodFM1, SodFM2 and SodFM4, we identified examples of cambialistic enzymes[12,18], and four subfamilies (SodFM1–4) had members that exhibited multiple different metal utilization preferences (Fig. 2a). We thus leveraged our large set of natural enzymes sampled from across the tree of life (Fig. 2a) to investigate if metal preference is a discrete three-state property (Fe, Mn or cambialistic), or rather forms a range of values between perfect Fe and Mn specificities. Although the term 'metal-specificity' is commonly used in reference to the disproportional activity of a SOD enzyme with Fe and Mn, the cambialistic SODs are inherently non-specific for either metal as they can use both[13]. Across the tested SodFMs, metal preferences, represented by approximate cambialism ratios (aCR, equal to the Fe-dependent activity divided by the Mn-dependent activity), formed a range of values (Fig. 2a–c) between higher Fe preference (aCR >2) and higher Mn preference (aCR <0.5), with perfect cambialism at the midpoint (aCR of 1), indicating that metal preference is best thought of as a continuum. We propose the term 'metal preference' instead of 'metal specificity' to better reflect the observation that most SodFMs are not completely metal specific. Note that the terms metal preference and metal specificity reflect the relative activity levels with alternative metal co-factors, and should not be confused with metal selectivity (metal-binding preference[19]), which refers to differences in the affinity of metal binding[20].

## Secondary sphere is key to SodFM metal-preference evolution

To test if aCR values matched with the identity of previously identified residues spatially located within the metal's secondary coordination sphere, $X_{D-2}$ (refs. 13,21,22) and $H_{Cterm}/Q_{Cterm}/Q_{Nterm}$[23–27] (Fig. 1c), we analysed the distribution of aCRs with respect to the residue found at these positions across SodFMs found in nature (Fig. 2). Higher Fe preference was usually associated with the presence of two alternative water-coordinating residues, $H_{Cterm}$ (69% FeSODs, 9/13 enzymes) or $Q_{Nterm}$ (72% FeSODs, 18/25 enzymes), whereas the enzymes containing $Q_{Cterm}$ were mostly Mn preferring (78% MnSODs, 21/27 enzymes). At the position $X_{D-2}$, the two smallest amino acids, glycine ($G_{D-2}$) or alanine ($A_{D-2}$) (Fig. 2b), were frequently found in SodFMs with higher Mn preference (Fig. 2c). More cambialistic outliers included the *Bacteroides fragilis* (aCR 0.98) and *Coprobacter fastidiosus* (aCR 0.99) SodFM2s, which seem to have evolved higher Mn activity from their most likely Fe-preferring ancestor. This confirmed that the presence of $G_{D-2}$ and $A_{D-2}$ within the context of otherwise predominantly Fe-preferring SodFMs can indicate potential evolutionary metal-preference modulation events. In contrast, either threonine ($T_{D-2}$) or valine ($V_{D-2}$) at this site (Fig. 2b) were usually found in SodFMs with higher Fe preference (Fig. 2c). The only two non-Fe-preferring outliers were two Bacteroidales SodFM1s from *Anaerophaga thermohalophila* ($T_{D-2}$; aCR 0.87) and Bacteroidetes RBG_13_43_22 ($T_{D-2}$; aCR 0.46). However, these two SodFM1s have probably evolved from highly Mn-preferring ancestors towards increased activity with Fe. This suggests that mutations of $X_{D-2}$ and $H_{Cterm}/Q_{Cterm}/Q_{Nterm}$ residues were involved in the identified evolutionary changes of metal preference.

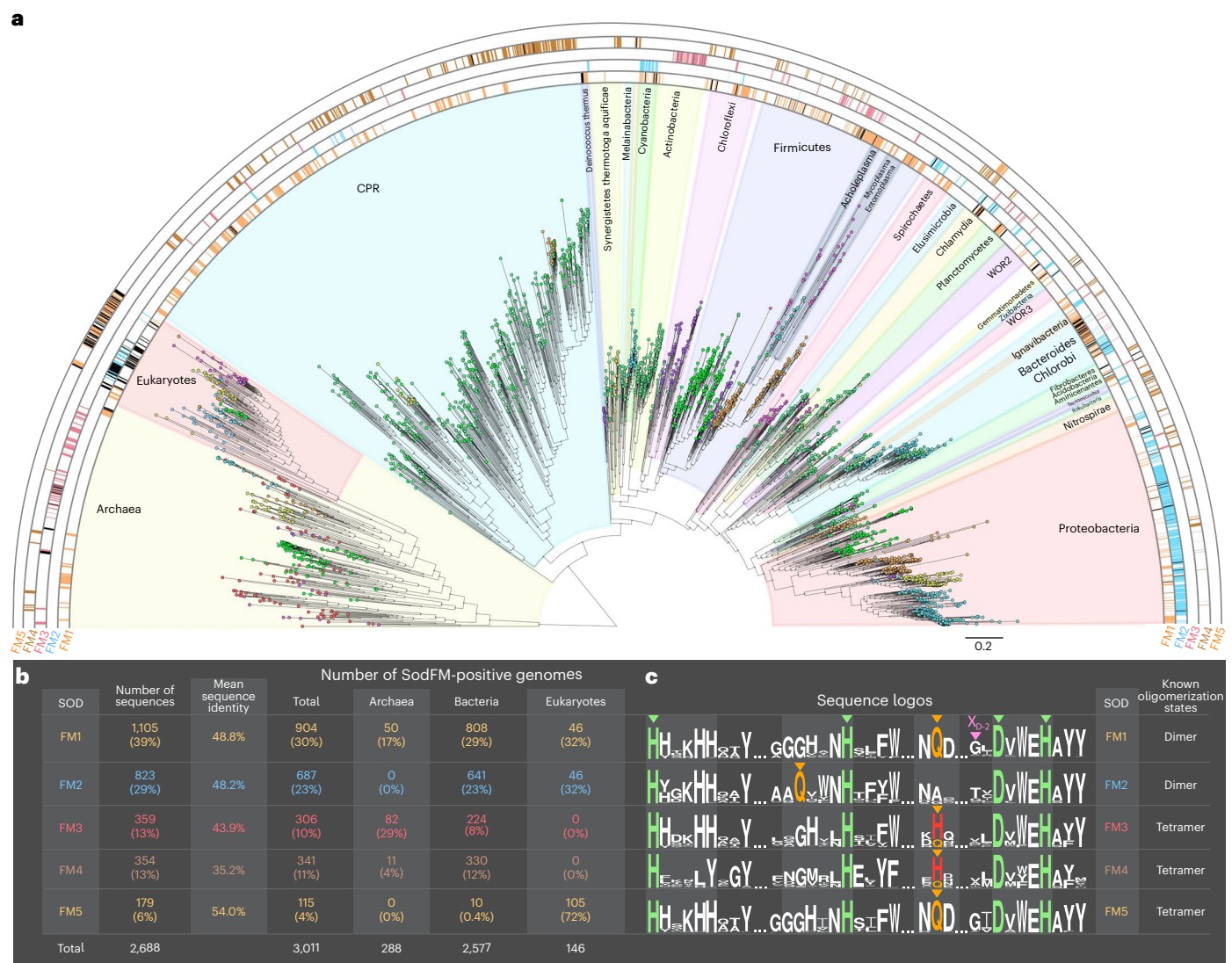

**Fig. 1 | Evolutionarily distinct SodFM subfamilies have wide and variable distribution across the tree of life. a**, Pattern of variable distribution of the five SodFM subfamilies (Extended Data Fig. 2) observed across the tree of life. The coloured semi-circles represent the distribution of SodFM1s (1,105 sequences, orange), SodFM2s (823 sequences, blue), SodFM3s (359 sequences, red), SodFM4s (354 sequences, brown) and SodFM5s (179 sequences, orange), mapped onto a phylogeny of 146 eukaryotes, 2,577 bacteria and 288 archaea. The presence of species with more than a single representative of a particular SodFM subfamily (black, Supplementary Data 18) reveals multiple independent lineage-specific SodFM1–5 subfamily expansions. **b**, Tables containing numbers and mean percentage of protein sequence identity of the SodFM1–5 subfamily members found within the analysis of 3,011 genomes sampled across the tree of life (**a**). **c**, Sequence logos illustrate frequencies of key residues found across the analysed

SodFMs, including four universally conserved metal-coordinating residues, three histidines and a single aspartate (green triangles). The key distinction among SodFM1–5 subfamilies is the identity and position of a water-coordinating residue (orange triangles) spatially located within the enzymes' catalytic centres: SodFM1/5s have a C-terminal Gln, SodFM2s have an N-terminal Gln, and most SodFM3–4s (75% sequences) have a C-terminal His and the rest of the SodFM3s and SodFM4s (25% each group) contained C-terminal Gln instead of His, often reflecting secondary His/Gln switches (Extended Data Fig. 2a). The key distinction between SodFM3s and SodFM4s is the loss of the conserved C-terminal His in the HXXXHH motif in SodFM4s. The key distinction between SodFM1s and SodFM5s is that the studied representatives of the former are homodimeric, whereas those of the latter are tetrameric (Extended Data Figs. 2a and 3a). Residue $X_{D-2}$ represents an important determinant of Fe/Mn metal preference (pink triangle).

Importantly, these isozymes also demonstrated that understanding an enzyme's evolutionary history can aid predictions of their properties.

## Switched metal preference evolved multiple independent times

To better understand the context of metal-preference evolution, we mapped the distribution of the identified residues and measured aCRs onto the trees of life (Fig. 3a–e and Extended Data Figs. 5 and 6). Clear patterns of metal-preference modulation were identified during the evolution of bacterial lineages important for human health including Bacteroidales (Fig. 3a and Extended Data Fig. 5), Mycobacteria

(Fig. 3d) and Staphylococci (Fig. 3e); and enigmatic uncultured Wolfe-bacteria (Fig. 3c) from bacterial candidate phyla radiations (CPR). In human-pathogenic Mycobacteria, acquisition of more Fe-preferring tetrameric SodFM3s has occurred at least twice (Fig. 3d), whereas a single neofunctionalization event could be identified in the ancestor of *Staphylococcus aureus* and *Staphylococcus argenteus* (Fig. 3e). The largely anaerobic Bacteroidales are a taxonomic group in which metal preferences have frequently switched. We identified at least two independent emergences of cambialism in this group, once from Fe-SodFM2 and a second time from Mn-SodFM1 ancestors. CPR Wolfebacteria represented a striking and highly unusual emergence of dimeric (Extended

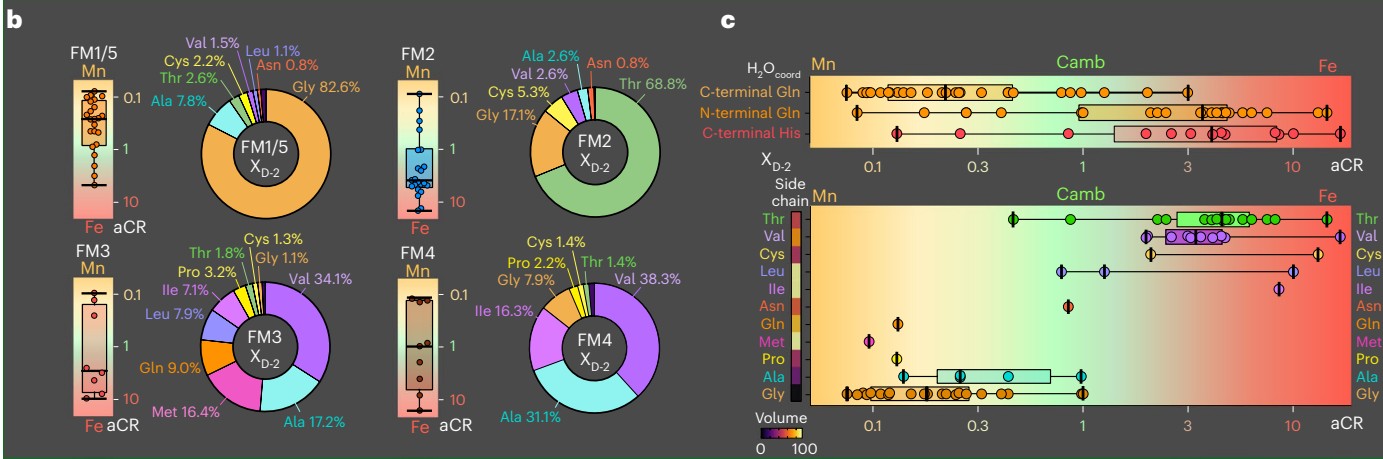

**Fig. 2 | Enzymes with diverse metal preferences can be found across the SodFM subfamilies. a**, The aCR values for the 64 characterized SodFMs were mapped onto the protein tree inferred from 2,688 SodFM sequences identified across the tree of life (Fig. 1a). A range of aCRs, from higher Fe preference (aCR >2, red), through cambialistic (0.5 < aCR < 2, green), to higher Mn preference (aCR <0.5, orange), were found across the SodFM subfamilies. Sensitivity to peroxide inhibition (+) was detected for Fe-preferring and Fe-loaded cambialistic enzymes, but not in the Mn-preferring enzymes, which were resistant to peroxide treatment (−). Identity of the metal's two key secondary coordination sphere residues, namely water-coordinating Gln/His and the $X_{D-2}$ residue located close to the Asp ligand (Fig. 1c), are displayed next to aCRs. **b**, Distribution of the aCRs for the SodFM subfamilies is presented as box and whiskers plots (left), and the frequencies of different amino acid residues found at the $X_{D-2}$ position

are presented as pie charts (right). SodFM1s and SodFM5s were grouped together here as they both contained C-terminal Gln and displayed similar metal preference and $X_{D-2}$ distribution patterns (SodFM5s $X_{D-2}$, G: 89%; A: 7%; T: 1.7%). The high proportion of Mn-preferring SodFM1s and Fe-preferring SodFM2s, and the predicted Gln/His and the $X_{D-2}$ residues in the reconstructed sequences of the ancestral nodes (**a**, dashed outlines), are consistent with the hypothesis that the LCAs of these two subfamilies were Mn and Fe preferring, respectively (Extended Data Fig. 2f,g). **c**, Distribution of the aCR values for SodFMs (top) with different water-coordinating residues (C-terminal Gln, N-terminal Gln or C-terminal His) and different $X_{D-2}$ residues (bottom) are displayed as box and whiskers plots on a logarithmic scale, colour coded to reflect apparent metal specificities (Mn: orange; cambialistic (camb): green; Fe: red). Amino acid residues were colour coded to enable comparisons among **a**, **b** and **c**.

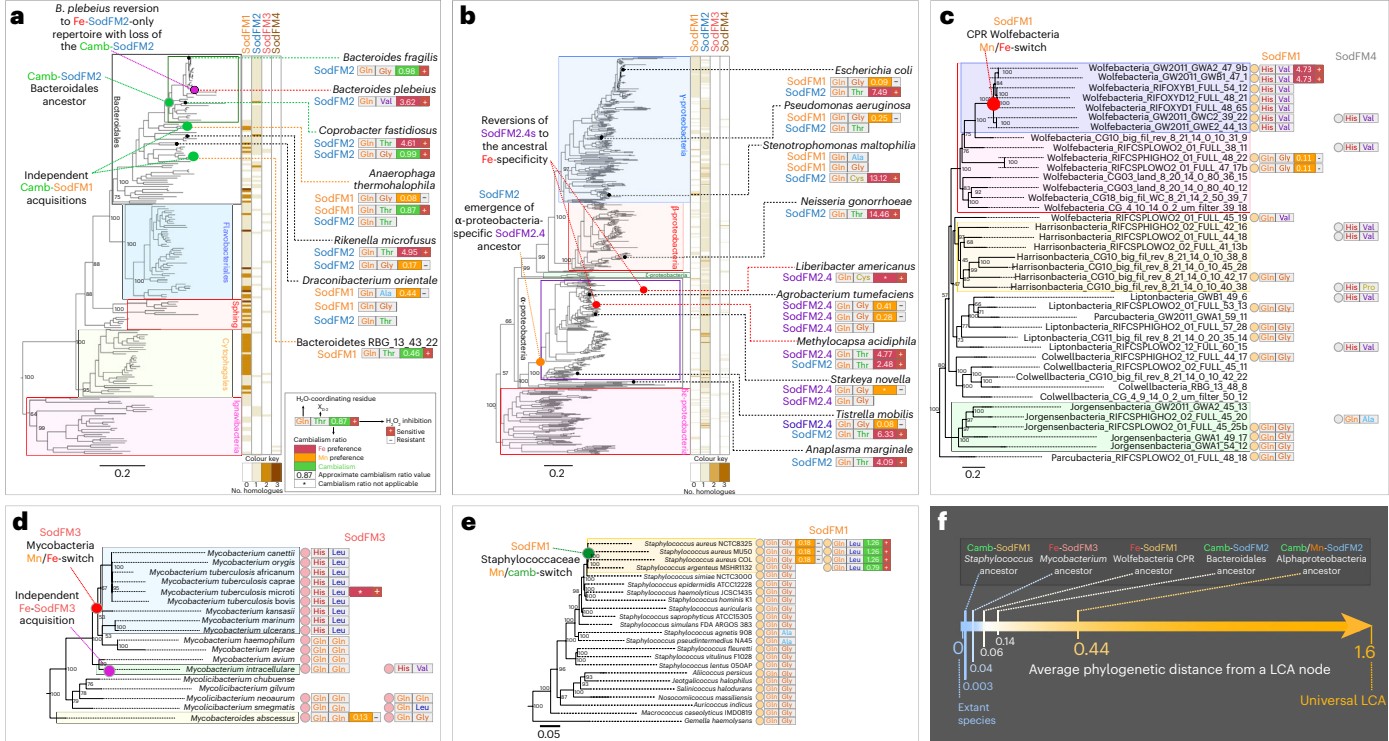

**Fig. 3 | SodFM metal preference and peroxide sensitivity have been changed independently many times within different evolutionary contexts.** **a–e** Distribution of SodFM1–4 subfamilies (orange–brown heat maps in **a** and **b**, or coloured circles in **c–e**) was mapped onto the species trees of Bacteroidota (**a**), Proteobacteria (**b**), CPR Wolfebacteria (**c**), Mycobacteriaceae (**d**) and Staphylococcaceae (**e**). In Gram-negative bacteria, SodFM repertoires consisted mainly of SodFM2s in Proteobacteria (**b**) and SodFM1s in Bacteroidota (**a**). Within the Bacteroidota, a subset of the Bacteroidales have switched to SodFM2 (**a**). SodFM1s and SodFM4s were found in uncultured CPR Wolfebacteria (**c**). In Gram-positive Mycobacteriaceae only SodFM3s were identified (**d**), whereas Staphylococcaceae contained only SodFM1s (**e**). The identities of the water-coordinating Gln/His, the residue $X_{D-2}$, and verified aCR and peroxide inhibition for the SodFMs characterized in this study were mapped onto the trees and annotated as previously described (Fig. 2). Inferred multiple independent

evolutionary shifts towards higher Fe preference and peroxide sensitivity (red circles), higher Mn preference and peroxide resistance (orange circles), emergences of cambialism (green circles) and likely SodFM repertoire changes via LGT (magenta circles) were annotated onto the trees (**a–e**). **f**, The average phylogenetic distances between the inferred LCA nodes (coloured circles in **a–e**) and their descendent tree tips were calculated and presented on a colour-coded scale, where the minimum value represents the extant homologues (tree tips) and the maximum value represents the average distances of all studied archaea, bacteria and eukaryotes from their LCA (universal LCA). Emergence of alphaproteobacterial SodFM2.4 was the most ancient and that of the cambialistic SodFM1 within staphylococci the most recent of the identified metal-preference evolutionary modulation events. Scale bars represent the number of substitutions per site.

Data Fig. 7) Fe-preferring SodFM1 containing SodFM3/4-like (Fig. 3c) $H_{Cterm}$ and $V_{D-2}$ residues (Fig. 1c). Importantly, our extensive analysis of SodFM metal utilization evolution provided further evidence demonstrating that the identity of $X_{D-2}$ and $H_{Cterm}/Q_{Cterm}/Q_{Nterm}$ combined with phylogenetic analysis can enable predictions of their metal preferences more accurate than those on the basis of sequence analysis or phylogenetics alone.

## Ancient SodFM metal-preference switch in Alphaproteobacteria

Based on predictions derived from our analysis, we identified a distinct group of Mn-preferring SodFM2s in Alphaproteobacteria (SodFM2.4; Fig. 3b and Extended Data Figs. 6 and 8) constituting a major component of marine and terrestrial ecosystems. On the basis of protein trees (Fig. 2a and Extended Data Fig. 6b), species trees (Fig. 3b and Extended Data Fig. 6a), amino acid correlation analysis (Extended Data Fig. 8) and distribution of the key amino acid residues (Extended Data Figs. 6b and 8), SodFM2.4s have evolved from ancestral Fe-preferring SodFM2 (Extended Data Figs. 6b and 8) in the LCA of all Alphaproteobacteria following the split from Rickettsiales (Extended Data Fig. 6a), and represent the most ancient identified evolutionary metal-preference switch (Fig. 3f) other than that at the split between SodFM1s and SodFM2s (Fig. 2a). Later, five divergent SodFM2.4 sequences reacquired more

typical SodFM2 residues, including $T/V/C_{D-2}$ in place of the SodFM2.4 hallmark residues $G_{D-2}$, and display Fe preference (for example, *Methylocapsa acidiphila*; Fig. 3b and Extended Data Fig. 6a), suggesting they have evolutionarily reverted to their ancestral metal preference on at least three independent occasions (Fig. 3b and Extended Data Figs. 6a,b and 8a). Extensive sequence analysis-driven mutagenesis of *Agrobacterium tumefaciens* SodFM2.4 (Extended Data Fig. 9m) revealed that, compared with changing from preferential utilization of a single metal co-factor to cambialism, as observed in *S. aureus* and Bacteroidetes (Fig. 3a,e), reversion from high Mn to high Fe preference may require more than a single mutation. It is therefore striking that SodFM2.4s from multiple species have evolved back to the ancestral Fe preference (Extended Data Fig. 6) rather than acquiring a canonical Fe-preferring SodFM2 via LGT.

## SodFM metal preference co-evolves with peroxide sensitivity

In addition to higher Mn preference, similarly to MnSodFM1s, MnSodFM2.4s were resistant to hydrogen peroxide inhibition (Extended Data Figs. 4c,k and 8a). It is well established that metal preference is strongly associated with the sensitivity of SodFMs to peroxide[28]. Mn-preferring and Mn-loaded cambialistic SODs tend to be resistant, whereas the activity of Fe-preferring and Fe-loaded cambialistic SODs

is typically inhibited by peroxide. Results from our characterization of isozymes sampled across the tree of life (Fig. 2a and Extended Data Fig. 4) showed that this biochemical feature was universally conserved across all studied SodFMs, consistent with it co-evolving with metal preference. Consequently, as in the case of metal preference, peroxide inhibition is not constrained to phylogenetic subfamilies, but has changed multiple independent times. This suggests that, in addition to adapting to changing metal bioavailability, metal preference could evolve in response to high concentrations of hydrogen peroxide, which is often used as a potent anti-microbial agent, both artificially in healthcare and by living organisms in nature.

## SodFMs vary in susceptibility to metal-preference modulation

To verify our hypothesis that SodFMs can evolve their metal preference and peroxide inhibition by mutating secondary coordination sphere residues $X_{D-2}$ and $H_{Cterm}/Q_{Cterm}/Q_{Nterm}$, we performed biochemical characterization of 50 synthetically generated mutants of the natural enzymes. Mutations were designed to alter metal preference by introducing residues from isozymes with opposing preference, guided by our analysis of the frequencies of the amino acids in these positions in natural enzymes (Fig. 2).

Of all tested mutants, 35 (70%) exhibited a significant shift in aCR value, indicating change in specific activity with one metal relative to that with the other metal. Of these 35 mutants, a total of 26 variants (52% of all mutants) displayed changes in the preferred metal relative to the wild type (WT) (Extended Data Fig. 9 and Supplementary Data 8), further supporting the significance of these residues in metal-preference determination. Decreased expression levels were observed for nine variants (18% of all mutants), suggesting that mutation of these residues can also have deleterious effects on protein stability and/or folding. No significant effects were observed in 15 (30%) mutants (Extended Data Fig. 9 and Supplementary Data 8), thus mutagenesis of $X_{D-2}$ and $H_{Cterm}/Q_{Cterm}/Q_{Nterm}$ does not always result in metal preference change. This indicates the importance of other regions of the protein fold for metal-preference modulation by $X_{D-2}$ and $H_{Cterm}/Q_{Cterm}/Q_{Nterm}$ mutagenesis. This is particularly evident in specialized metal-preferring SODs such as *Escherichia coli* SodA, which does not acquire Fe when homologously overexpressed in our *E. coli* strain in aerobic conditions (Extended Data Fig. 10f); and in *Bacillus anthracis* SodA2 (Fe-preferring SodFM1) or *A. tumefaciens* Mn-preferring SodFM2.4, where mutagenesis of $X_{D-2}$ had no effect on their metal preference, although it played a clear role in the metal preference switch of their ancestor. This suggests that, following the metal-preference switch, subsequent changes in other regions of the fold can de-correlate later effects of the $X_{D-2}$ and $H_{Cterm}/Q_{Cterm}/Q_{Nterm}$ mutations from their initial effects, a process termed epistatic drift[29]. Alternatively, metal-preference changes via $X_{D-2}$ and $H_{Cterm}/Q_{Cterm}/Q_{Nterm}$ in some SodFMs could require preceding changes in other regions of the fold (intramolecular pre-adaptations).

Regardless of the underlying evolutionary mechanisms, the residues $X_{D-2}$ and $H_{Cterm}/Q_{Cterm}/Q_{Nterm}$ constitute a strong indicator of metal preference in natural enzymes (Fig. 2). Although the susceptibility of SodFMs to metal-preference modulation by mutating these residues was not universal, the majority of variants in which these residues were mutated exhibited significant modulation of metal utilization.

## Cambialism can be a positively selected property in nature

Next, we leveraged the cambialistic SODs from *S. aureus* and *B. fragilis* to investigate the specific role of the $X_{D-2}$ position in determining metal preference (Fig. 4a,b and Extended Data Fig. 9a–d). We investigated multiple variants in which their $X_{D-2}$ residues were mutated (Extended Data Fig. 9a–d), guided by our analysis of which amino acid appears at this position in natural SodFM sequences (Fig. 2b). In the *S. aureus* cambialistic SodFM1, the magnitude of aCR change was dependent on

which amino acid was introduced at $X_{D-2}$ (Fig. 4a), and was consistent with the correlation between the aCR and $X_{D-2}$ residues of SodFMs sampled across the tree of life (Fig. 2c). Importantly, while three *S. aureus* mutants shifted back towards the ancestral Mn preference, three mutants showed greater Fe preference (Fig. 4a). The shift towards Fe preference was observed for the two amino acids (Thr and Val) most often present in Fe-preferring SODs, while a strong change towards the ancestral Mn specificity was observed for the two amino acids (Ala and Gly) most commonly found in Mn-preferring SODs. This capacity for bidirectionally changing metal preference in *S. aureus* SodM was unique across the tested SODs, suggesting it can provide a powerful model system for future studies of how metal preference is determined. This contrasted with the *B. fragilis* cambialistic SodFM2 (Fig. 4b and Extended Data Fig. 9c,d) where the majority of mutants, regardless of the amino acid introduced at the $X_{D-2}$ position, were shifted towards the ancestral Fe preference. This can probably be explained by the structure of the amino acid (glycine) found in *B. fragilis* $X_{D-2}$ position (Fig. 4e), which has no sidechain, thus replacement with any sidechain found in other amino acids affects its local environment and influences metal preference (Fig. 4b). Therefore, our results suggest that the *S. aureus* cambialistic SodFM1 has potential for evolution towards higher Fe preference, whereas the *B. fragilis* cambialistic SodFM2 has reached the highest possible level of Mn preference via mutagenesis of the $X_{D-2}$ residue alone. Crucially, the cambialistic sequence is conserved in all analysed *S. aureus* and *S. argenteus* genomes, despite our observation that only a single $X_{D-2}$ mutation is required to switch *S. aureus* cambialistic SodFM1 into an Fe-preferring SOD with substantially increased Fe-dependent activity, providing evidence for the selection of cambialism rather than Fe preference in *S. aureus*. We thus propose that cambialism need not strictly be an evolutionary intermediate stage between two metal specificities but rather the desired property that can be selected for in nature, probably in this case for circumventing nutritional immunity[12].

## Multiple evolutionary pathways can modulate metal preference

To further test the wider evolutionary significance of the secondary coordination sphere residues, we investigated double mutant variants combining mutations at positions $X_{D-2}$ and the water-coordinating $Q/H_{Cterm}$. Our characterization of these variants revealed multiple evolutionary pathways towards metal-preference switching. In some cases, such as *Nesterenkonia alba* SodFM3 Q/H, a single residue has a major impact on the preferred metal co-factor (Fig. 4g and Extended Data Fig. 9k,l), while in others, such as CPR SodFM1 H/Q, mutation at the same position had a negligible effect (Fig. 4c and Extended Data Fig. 9e,f). In some, such as CPR SodFM1 H/Q $V_{D-2}$G, only a cumulative effect of the two mutations affected the metal-preference (Fig. 4c and Extended Data Fig. 9e,f), whereas no significant effect was observed in either single or double mutants of nanoarchaeon *Nanopusillus acidilobi* SodFM4 (Extended Data Fig. 9i,j). The latter observation suggests that additional loci in the SodFM sequences must also regulate metal preference, which await identification. SodFM metal preference is probably modulated via fine tuning of the metal co-factor's redox potential[13,26,27,30]. Interestingly, in all $X_{D-2}$ and $Q/H_{Cterm}$ mutants with significantly changed metal preference, activity with at least one metal decreased (Fig. 4i). Significantly increased activity with both metals at the same time was not observed in any of these mutants. This suggests that a substantial and concomitant increase in activity with both metals via $X_{D-2}$ and $H_{Cterm}/Q_{Cterm}/Q_{Nterm}$ mutagenesis is impossible due to biophysical constraints. This is consistent with the 'redox tuning' hypothesis[31] under the assumption that structural changes that modulate the reduction potential of one metal should also change the reduction potential of the alternative metal within the same active site. This also indicates that SodFMs have evolved to maximize their metal-specific activity via $X_{D-2}$ and Q/H mutagenesis, but within limitations enforced

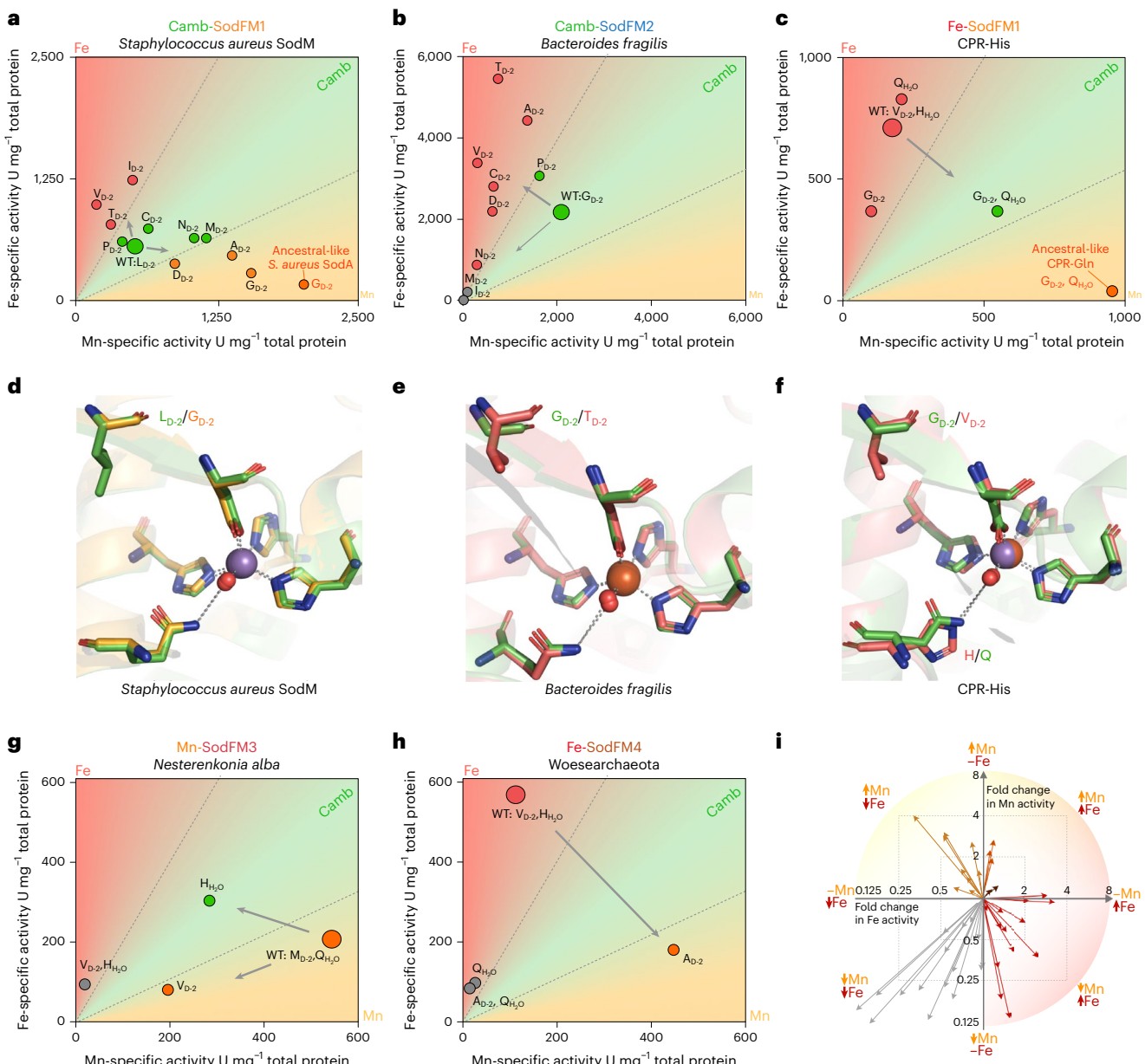

**Fig. 4 | Functional studies reveal multiple molecular pathways to metal-preference modulation in SodFM metalloenzymes. a,b,** Residues at the $X_{D-2}$ position in cambialistic *S. aureus* SodFM1 ($L_{D-2}$, **a**) and *B. fragilis* SodFM2 ($G_{D-2}$, **b**) were mutated to those found commonly at this position in natural SodFMs (Fig. 2b). The average specific activities of the WTs and mutants loaded with either Mn (*x* axis) or Fe (*y* axis) were plotted as colour-coded circles. In *S. aureus* (**a**), residues most commonly found in Mn-preferring SODs, $A_{D-2}$ and $G_{D-2}$ (Fig. 2a,b), shifted metal preference towards Mn (orange circles), whereas those found mainly in Fe-preferring SODs, $V_{D-2}$ and $T_{D-2}$ (Fig. 2a,b), shifted towards higher Fe preference (red circles). In the cambialistic SodFM2 from *B. fragilis* (**b**), all mutants displayed lower activity with Mn and the majority had higher activity with Fe, representing change towards its most likely ancestral Fe-preferring phenotype. **c,** For Fe-preferring CPR Wolfebacteria SodFM1 (CPR-His), mutations in both the $X_{D-2}$ and water-coordinating H/Q residues were necessary to shift towards the inferred ancestral higher Mn activity. The likely ancestral phenotype was proxied by the closely related SodFM1 (Fig. 2a) from more basal Wolfebacteria (Fig. 3c). **g,h,** Single mutants, but not double

mutants, shifted metal preference of *N. alba* SodFM3 (**g**, Q/H mutant) and Woesearchaeota SodFM4 (**h**, $V_{D-2}$A). **d–f,** Crystal structures of *S. aureus* SodFM1 $L_{D-2}$G (**d**), *B. fragilis* SodFM2 $G_{D-2}$T (**e**) and CPR-His SodFM1 $V_{D-2}$G H/Q double mutant (**f**) were analyzed to investigate structural effects of these mutations. The structures in **e** and **f** were solved in this study. Cartoon representation of the active sites were superimposed onto corresponding WT. Apart from the mutated residues, there were no significant changes to the position of metal (manganese, blue, and iron, orange spheres), catalytic water molecules (red spheres) and primary metal coordination bonds (dashed lines) within the resolution limit of the solved structures (backbone root mean squared deviation (RMSD) of 0.4 Å, and metal primary coordination sphere RMSD below 0.1 Å). **i,** Plot represents fold changes in activity with Mn and Fe of the tested SodFM mutants relative to the WTs (Supplementary Data 8), where arrows indicate significant increase in activity with only one metal (Fe, red; Mn, orange), or decline in activity with both metals (grey). Neither of the increases in activity with both metals in a single mutant (brown) were significant. The fold changes in activity are represented on a log2 scale.

by likely epistatic effects[32] of other sites within their protein structure. Importantly, consistent with previous reports[13,26,27,33], comparison of four protein crystal structures of the WT and mutant SodFMs we

determined here (Fig. 4e,f and Extended Data Fig. 7) indicated that metal-preference modulation by $X_{D-2}$ and Q/H occurs without any significant changes to the secondary metal coordination sphere within

the resolution limits of the solved structures. This provides further support for 'redox tuning' via subtle structural changes[13,34]. Better understanding of the residues that determine metal preference and its evolvability[35] in SodFMs can enable development of models describing how evolutionary changes in metalloprotein structure can shape their biophysical properties.

## Discussion

Despite substantial and continuing progress in our understanding of protein functions and the evolutionary processes that shape them, a vast gap exists between the number of known natural protein sequences and the number of those that have been investigated experimentally. Bridging this gap is required to further our understanding of the fundamental mechanisms involved in the functioning and evolution of all living systems, and to apply this knowledge in medicine, biotechnology and synthetic biology.

Although SodFMs are a well-studied protein family, with over 50 WT protein crystal structures available from various organisms, we were able to uncover additional diversity via application of bioinformatics and phylogenetics on a dataset of genomes sampled across the tree of life. This included the novel categorization of the SodFM protein family into five distinct subfamilies, the first comparison of multiple members of the largely unstudied group of SodFM4s, and identification of atypical isozymes such as the Fe-preferring SodFM1 from CPR Wolfebacterium. It also enabled us to identify multiple independent evolutionary events, both ancient and more recent, in which their key biophysical property of Fe/Mn metal preference was switched. *S. aureus* cambialistic SodFM1, involved in evasion of the mammalian immune response[12,13], has emerged from the most recent of these switches, and is susceptible to modulation of its metal preference in both directions. This contrasted with the Agrobacterium Mn-preferring SodFM2.4, which descended from the most ancient identified metal-preference switch, where modulation via mutagenesis of the identified key residues was no longer possible. This is probably due to epistatic drift, which can de-correlate the later effects of mutations from their initial effects[29]. We also demonstrated that Agrobacterium Mn-SodFM2.4, and all other tested SodFMs active with Mn, are resistant to hydrogen peroxide inhibition, which could provide additional selective advantage, for example, to organisms involved in peroxide scavenging within microbial communities[36–38].

Metal preferences of SodFMs are commonly misannotated in databases and cannot be predicted by sequence analysis or phylogenetics alone, but our application of both methodologies in combination improved prediction accuracy. Many other protein families include metalloproteins, and although metal preference has been established for some members of some families, its diversity has probably been frequently underappreciated. For example, the unexpected discovery of copper-only SODs, devoid of the requirement for zinc, among the Cu/Zn-dependent SOD family requires reassessment of the annotation of these distinct superoxide detoxification enzymes[39,40]. Similarly, a carbonic anhydrase isozyme from marine diatoms has evolved to utilize cadmium, in place of the usual Zn, as an adaptation to oceanic Zn deficiency[41]. In SodFMs, metal preference can be changed within short evolutionary timeframes. Although the selection pressures driving their evolution remain to be evaluated, we hypothesize that environmental conditions affecting intracellular metal homeostasis are probably frequent drivers. The same selection pressures would affect other metalloproteins, making the groups of organisms with signatures of evolutionary SodFM metal preferences changes, such as Alphaproteobacteria and Bacteroides, prime candidates for investigations of metal-preference evolution in other protein families. This could have particularly important implications in healthcare as metal homeostasis and correct protein metalation are important in human disease[9] and in microbial infections[6], and have been suggested to be promising anti-microbial therapeutic targets[10]. Furthermore, the ability

to engineer cambialistic synthetic proteins can enable the utilization of different metals to perform the same activity, probably providing economic and environmental benefits in biotechnological applications.

The existence of Fe-preferring, Mn-preferring and cambialistic SodFMs is well established, and their metal-preference changes were previously hypothesized as being related to ancient geological changes such as Earth's atmosphere oxygenation[11,14] and adaptation of pathogens to nutritional immunity[12,13]. Here we identified multiple additional metal-preference changes and demonstrated that, rather than switching between three discrete states of metal specificity, SodFM metal preference evolves by sliding on a continuous scale of cambialism ratio (CR). This evolutionary fine tuning probably occurs through the proposed biophysical mechanism of redox tuning[31], a hypothesis supported by our biochemical and structural analysis of natural SodFMs and their metal-preference switched mutants. It was previously hypothesized that ancient ancestral SodFMs were cambialistic[14,42], but in light of our data, we hypothesize that the ancient SodFMs would have been susceptible to metal-preference modulation, similarly to the extant homologues. A physiological role has only been previously demonstrated for one cambialistic SodFM, in *S. aureus* pathogenesis. Yet our data also indicate that on modern Earth, cambialism can be more common and more important than previously anticipated as, for example, in the Bacteroidales, cambialism has evolved multiple independent times from different ancestral backgrounds. However, the wider biological significance and frequency of cambialism and evolutionary metal-preference modulation in other protein families remains to be explored. Metals are indispensable for modern cellular life, probably played important catalytic functions at its origin, and their bioavailability has changed dramatically over the Earth's history. However, it is not always clear why one metal was selected over another to play a particular biological role or whether the selected metal can be easily changed for another metal with similar biophysical properties in the course of evolution. Our findings suggest that in the case of SodFMs, the preference for one of the two possible metal co-factors can and has been modulated multiple independent times in the course of the SodFM protein family's evolution, and remains susceptible to further changes in the extant family members.

## Methods

### Generation of Δ*sodA*Δ*sodB* expression strains of *E. coli* BL21(DE3) and pLysSRARE2 (DE3) pLysS

The *sodFM* gene knockout (KO) strain of *E. coli* BL21(λ DE3) was generated following a standard protocol[43] with minor modifications. First, the sequences of *sodA* and *sodB* genes in the BL21 genome (GCF_000022665.1, ASM2266v1) were compared with that of the *E. coli* K-12 strain (GCF_000005845.2, ASM584v2) where successful gene KO strains were previously generated[44]. Specificity of designed primers with homology to the targeted genes was verified using local BlastN (-task blastn-short, e-value 0.1)[45] against the whole BL21 genome. *E. coli* BL21(λDE3) cells expressing λ Red recombinase (BL21-RED) from pKD46 plasmid were induced with 10 mM arabinose at $OD_{600}$ of 0.2 and made electrocompetent at $OD_{600}$ of 0.6 by three washes in ice cold 10% (vol/vol) glycerol. A kanamycin resistance cassette with recombination overhangs was amplified from pKD4 plasmid template with Q5 Hot Start DNA Polymerase (NEB), purified with Monarch DNA gel extraction kit (NEB), and electroporated (500 ng DNA, 250 ng μl$^{-1}$) into 90 μl electrocompetent BL21-RED cells at 2.5 kV, 200 Ohm, 0.025 F for 4.5 ms (BTX ECM 630) in 0.1 cm electroporation cuvettes (VWR), followed by 2 h recovery in super optimal broth with catabolite repression (SOC) medium. All bacterial cultures (except recovery following electroporation in SOC medium) were grown in Luria-Bertani medium supplemented with 0.2% (wt/vol) glucose, with 1.5% (wt/vol) agar for solid medium. The remaining steps of the protocol were performed as described in the original publication[43]. The BL21(λDE3) Δ*sodB* mutant was used for subsequent generation of the BL21(λDE3) Δ*sodA*Δ*sodB*

double mutant following the same protocol. Correct antibiotic resistance cassette insertions were verified using diagnostic polymerase chain reaction (PCR) with OneTaq Quick-Load 2× Master Mix (NEB). The gene deletion was verified using PCR with sodA(B)_screen_F(R) primers complementary to regions upstream and downstream from the deleted loci, and its phenotype confirmed using the in-gel SOD activity assay on soluble protein extracts from the WT and KO strains (Extended Data Fig. 10c). The BL21(DE3) pLysSRARE2 Δ*sodA*Δ*sodB* strain was constructed by transforming the BL21(λDE3) Δ*sodA*Δ*sodB* with pLysSRARE2 plasmid isolated from the commercially available ROSETTA2 (λDE3) pLysS strain (Novagen). All primers used for KO strain generation and verification were synthesized by Eurofins Genomics, with sequences provided in Supplementary Data 1.

## SOD genes synthesis, cloning and site-directed mutagenesis

Amino acid sequences of SodFMs of interest identified in bioinformatic analyses were used to generate gene sequences codon optimized for expression in *Escherichia coli* B using commercial Codon Optimization Tool (IDT). Predicted N-terminal targeting sequences identified with SignalP5 (ref. [46]), TargetP 2.0 (ref. [47]), ChloroP[48] and Phobius[49], and fusion domains identified with InterProScan 5 (ref. [50]), were manually verified in protein alignments as not being a part of the structural and functional conserved core of the SOD fold, and were excluded from the construct sequences. Sequence extensions complementary to the cloning site within pET-22b(+) (Novagen) were added to each codon-optimized gene and resulting constructs (Supplementary Data 2) were synthesized as gBlocks gene fragments (IDT). The amino acid sequences of the 82 WT SOD constructs investigated in this study are listed in Supplementary Data 3. The synthetic constructs were used directly for enhanced Gibson assembly cloning[51,52] into the pET-22b expression vector backbone. Ratios of 4 fmol plasmid (15 ng): 24 fmol insert (usually 10 ng) in DNA mix, and 2.5 μl of the DNA mix: 7.5 μl enzyme mix were used for the enhanced Gibson assembly. Site-directed mutagenesis of the SodFMs in pET-22b was performed using Q5 Hot Start DNA Polymerase (NEB) and the KLD kit (NEB) following the manufacturer's protocol with the single exception of using 5 μl instead of 10 μl final reaction volume. Sequences of all primers used in site-directed mutagenesis are provided in Supplementary Data 1. All plasmid propagations steps were performed in *E. coli* DH5α.

## Expressing SodFMs in the presence of Fe or Mn in 24-well culture plates

For screening of soluble expression and aCR verification experiments, *E. coli* BL21(λDE3) Δ*sodA*Δ*sodB* cells transformed with pET-22b SOD expression constructs were cultured in 4–5 ml M9 medium (0.1 mM CaCl₂, 2 mM MgSO₄, 0.4% glucose, 0.2% casamino acids, 1 mM thiamine, 12.8 g l⁻¹ Na₂HPO₄ × 7H₂O, 3 g l⁻¹ KH₂PO₄, 0.5 g l⁻¹ NaCl and 1 g l⁻¹ NH₄Cl) in 24-well culture plates (Whatman Uniplate) at 37 °C with 180 r.p.m. orbital shaking. Pre-cultures were grown for 16 h in M9 medium containing 100 μg ml⁻¹ ampicillin without metal supplementation. Aliquots (200 μl) of the pre-cultures were used to inoculate 4 ml M9 medium with ampicillin (100 μg ml⁻¹) supplemented with either 500 μM MnCl₂ or 100 μM Fe(NH₄)₂SO₄ and cultured until OD₆₀₀ reached 0.4–0.6. At that point, additional Mn (1 mM final concentration) or Fe (200 μM final concentration) and isopropyl β-ᴅ-1-thiogalactopyranoside (1 mM final concentration) were added, and culture was continued for 4 h. Cultures in Fe- or Mn-supplemented media were always performed in separate 24-well plates to avoid metal cross-contamination. The final OD₆₀₀ was measured, and the 24-well culture plates were centrifuged for 10 min at 4,000*g*. Spent media were removed, and cell pellets were resuspended in cell lysis solution (20 mM Tris pH 7.5, 100 μg ml⁻¹ lysozyme, 10 μg ml⁻¹ DNAse I, 1× Roche cOmplete ethylenediaminetetraacetic acid (EDTA)-free protease inhibitors), volume normalized on the basis of their final OD₆₀₀ measurements and transferred to 1.5 ml Eppendorf tubes. *E. coli* cells were

lysed by 10 s sonication (MSE Soniprep 150 with exponential probe at 10–14 μm amplitude) in a cold room and centrifuged for 20 min at 21,100*g*. The same protocol was followed for BL21(DE3) pLysSRARE2 Δ*sodA*Δ*sodB* except for 16 h expression at 16 °C in M9 media containing 100 μg ml⁻¹ ampicillin and 30 μg ml⁻¹ chloramphenicol. The supernatants (soluble protein extracts) containing the expressed SodFMs were used in the subsequent bicinchoninic acid protein concentration assay (Pierce), liquid and in-gel SOD activity assays, and sodium dodecyl sulfate–polyacrylamide gel electrophoresis (SDS–PAGE). Metal purity of culture media were verified using inductively coupled plasma mass spectrometry (ICP–MS).

## Protein expression and purification

Proteins were expressed in *E. coli* BL21(DE3) Δ*sodA*Δ*sodB* in 1 l M9 medium supplemented with 200 μM Fe(NH₄)₂SO₄ or 1 mM MnCl₂ and induced with 1 mM isopropyl β-ᴅ-1-thiogalactopyranoside for 4 h at 37 °C with 220 r.p.m. orbital shaking in 2 l baffled flasks. Bacterial cells were lysed by sonication of washed cell pellets in 20 mM Tris pH 7.5, 1× cOmplete EDTA-free protease inhibitor (Roche), 100 μg ml⁻¹ lysozyme, 10 μg ml⁻¹ DNAse I, followed by centrifugation at 19,000 *g* at 4 °C. Cleared cell lysates were subjected to chromatographic separation using an ÄKTA purification system (Cytiva). Standard protocol for most characterized recombinant SOD proteins consisted of purification using anion exchange chromatography (HiTrap Q HP column, Cytiva) in 20 mM Tris pH 7.5 buffer with 0–1 M NaCl linear gradient elution, and subsequent size exclusion chromatography in 20 mM Tris pH 7.5, 150 mM NaCl buffer on a Superdex 200 16/600 (Cytiva) column. For purification of *B. fragilis* SodFM2 and its mutants, 20 mM Tris pH 8.5 was used. The *E. coli* SodFM1 (SodA) was collected in the flowthrough during anion exchange chromatography and was subsequently subjected to additional cation exchange chromatography (5 ml SP FF column (Cytiva) in 20 mM MES pH 5.5, 0–1 M NaCl linear gradient elution), where it was also collected in the flowthrough. Metal content of protein extraction buffers as well as metal loading of the purified protein preparations were verified using ICP–MS.

Analytical size exclusion chromatography using a Superdex 200 10/300 increase column (Cytiva) was performed to estimate the molecular weight of protein preparations in non-denaturing conditions based on elution volume. The column was calibrated using a mixture of molecular weight standards ranging from 1,350 to 670,000 Da (Bio-Rad) and blue dextran (2,000 kDa, Sigma) in 20 mM Tris pH 7.5, 150 mM NaCl buffer.

## SOD activity assays and (a)CR calculations

SOD activity was assessed quantitatively using the riboflavin-nitro blue tetrazolium (NBT) liquid assay[53] with minor modifications. The assay was performed in 96-well plates, in which 20 μl of a sample was mixed in each well with 180 μl assay solution (10 mM methionine, 1.4 μM riboflavin, 66 μM NBT and 10 μM EDTA in 50 mM potassium phosphate buffer, pH 7.8), immediately before 10 min incubation on a white light box followed by immediate measurement of absorbance at 560 nm. Serial twofold dilutions of tested samples were assayed to identify enzyme concentrations within the linear range of the assay that were close to 1 U in reference to a standard curve obtained using commercial bovine SOD standard (S5639, Sigma). To calculate specific SOD activities, the assay was performed in triplicate using samples diluted to the concentrations within the assay's linear range. CRs were calculated as a ratio of SOD activities of the verified Fe-loaded protein preparation divided by that of the verified Mn-loaded protein preparation. aCRs were calculated the same way but using SOD activity values of soluble protein extract samples from the 24-well culture experiments after lysis and centrifugation. CR values >0.5 were considered as evidence of higher Fe preference, CR <0.5 as evidence of higher Mn preference and enzymes with 0.5 < CRs < 2 were considered cambialistic.

For qualitative SOD activity assessment, approximately 15 µg (determined by bicinchoninic acid assay) of soluble protein extracts were separated by 15% (wt/vol) acrylamide native polyacrylamide gel electrophoresis (PAGE) and assayed using an in-gel assay as described previously[13]. The in-gel peroxide inhibition assay was performed as described previously[13]. For protein staining, samples were resolved on 15% (wt/vol) acrylamide SDS-PAGE gels and stained with Quick Coomassie (Protein Ark). All gels were imaged using a ChemiDoc imaging system (Bio-Rad), using the same settings (including exposure time) for all gels compared within a single experiment. Band intensity was quantified with Image Lab 6.1 software (Bio-Rad) using a rectangle volume tool to outline activity bands with local background subtraction. aCRs from the in-gel assay were calculated using the measured band intensities following the formula: $aCR = (intFe - intFeH_2O_2)/intMnH_2O_2$, where $intFeH_2O_2$ is band intensity of the Fe-loaded form following peroxide inhibition, intFe refers to the band intensity of Fe samples in non-peroxide-treated gels, and $intMnH_2O_2$ is the band intensity of the Mn-loaded form following peroxide inhibition. Data available in Supplementary Data 8, 9 and 10.

## Metal concentration verification using ICP–MS

Aliquots of protein extraction buffers or purified protein samples (20 µM) in 20 mM Tris pH 7.5, 150 mM NaCl were each diluted to 5 ml with 2% $HNO_3$ for elemental analysis. Elemental composition of the resulting acid solutions were quantified using an iCAP RQ ICP–MS instrument (Thermo Fisher Scientific) according to the manufacturer's specifications through comparison with matrix-matched elemental standard solutions, using In and Ir (10 µg l$^{-1}$) as internal standards (University of Plymouth Enterprise Ltd).

## Protein crystallography

Purified and concentrated protein samples were subjected to crystallization with commercially available matrix screens: PACT, JCSG+, Structure, Morpheus (Molecular Dimensions) and Index (Hampton Research) in 96-well Swissci MRC crystallization plates (Molecular Dimensions) using a Mosquito liquid handling robot (TTP Labtech), with the sitting drop vapour-diffusion method at 20 °C. CPR-His Wolfebacteria SodFM1 crystallized in 100 mM sodium acetate trihydrate pH 4.5 and 25% wt/vol polyethylene glycol 3350; CPR-His Wolfebacteria SodFM1 mutant crystallized in 200 mM sodium chloride, 100 mM sodium acetate pH 5.0 and 20% wt/vol polyethylene glycol 6000; B. fragilis SodFM2 crystallized in 200 mM sodium chloride, 100 mM Tris pH 8.5 and 25% wt/vol polyethylene glycol 3350; and B. fragilis SodFM2 mutant crystallized in 10 mM zinc chloride, 100 mM sodium acetate pH 5.0 and 20% wt/vol polyethylene glycol 6000. All crystals were cryo-protected with the addition of 20% polyethylene glycol 400 to the crystallization condition. Diffraction data were collected on the I03 and I24 beamlines at the Diamond Light Source Synchrotron (Didcot, UK) at 100 K. Data processing and refinement statistics are given in Supplementary Data 16. Data were indexed and integrated with the pipeline Xia2 (ref. 54) with either 3dii XDS[55] or DIALS[56] and scaled with Aimless[57]. Space group determination was confirmed with Pointless[58]. Five per cent of observations were randomly selected for the Rfree set. The phase problem was solved by molecular replacement using Phaser[59]. The 1UES was used as a search model in molecular replacement for B. fragilis SodFM2 structures, whereas 1IX9 and 1GV3 were used as search models in molecular replacement for CPR-His SodFM1 and its mutant, respectively. Initial model building was done with CCP4build task on CCP4cloud[60]. The model were refined with Refmac[61] and manual model building was done with Coot[62] (Supplementary Data 16). Models were validated with Molprobity[63] and Coot. Structural representations were prepared with PyMOL (Schrödinger, LLC).

Structural comparison was performed for global secondary structure matching in Coot and for least square comparison (LSQ-kab) of metal-binding residues in CCP4i. To visualize and compare electron density between structures, the 2Fo-Fc map was displayed at 1σ level.

## Bioinformatic protein family identification and characterization

Our previously described dataset of genomes sampled across the tree of life[13] was expanded with additional Bacteroidales and Alphaproteobacteria genomes to improve sampling of the lineages identified as metal-preference switching hotspots. The final dataset contained 3,058 genomes including 2,613 bacteria, 281 archaea and 146 eukaryotes (full list in Supplementary Data 4). Members of the SOD protein families were identified using PFAM[64] hidden Markov model (HMM) profiles (Sod_Fe_C, PF02777.21; Sod_Fe_N, PF00081.25; Sod_Cu, PF00080.23; SodNi, PF09055.14) as queries against a local database of 10,304,216 protein sequences encoded within all 3,058 analysed genomes in hmmsearch (-E 1e-5) profile search implemented in HMMER3.3 (ref. 65). Superoxide reductase (SOR) protein family members were identified with hmmersearch using PFAM[64] profiles (Ferric_reduct, PF01794; and Desulfoferrodox, PF01880), as well as BlastP (e-value threshold of 0.005)[45] searches using sequences of structurally characterized SORs (protein data bank (PDB): 1DO6, and 1DFX). Data available in Supplementary Data 17.

The identified 2,820 SodFM homologues (including 116 sequences with solved crystal structures available in the PDB) were aligned in MAFFT[66] with a set gap opening threshold (-op 2), and trimmed with trimAL[67] with a set gap threshold (-gt 0.1) to a final alignment length of 244 positions (Supplementary Data 5). The SodFM phylogeny (Supplementary Data 6) was generated with 1,000 ultrafast bootstrap[68] replicates in IQ-TREE[69], with the best fitting phylogenetic model WAG + R10 selected with ModelFinder[70] according to Bayesian information criterion implemented in IQ-TREE. Available solved SodFM crystal structures (listed in Supplementary Data 11–15) were downloaded from the PDB;[71] aligned, manually inspected and displayed in PyMOL (Schrödinger, LLC.); protein 3D structural comparisons were performed using the 'All against all' function implemented in DaliLiteV5.1 on a local workstation[72]. Groups of correlated amino acid residues within SodFM alignments were identified using an amino acid correlation algorithm implemented in PFstats[73].

## Generation of the tree of life

HMM profiles were generated from a curated set of 21 single-copy orthologues shared between bacteria, archaea and eukaryotes[74]. Sequences of the orthologues were downloaded from the supplementary material of the previous study of the universally conserved orthologues[74], aligned with MAFFT[66], trimmed with trimAL[67] (gappyout mode) and used for the generation of HMM profiles in HMMER3.3[65]. The profiles were subsequently used in hmmsearch (-E 1e-5) profile search[65] against the local database of 10,304,216 protein sequences encoded within 3,058 genomes sampled from across the three domains of life. Each of the identified orthologue groups were aligned with MAFFT[66], trimmed with trimAL[67] (gappyout mode) and used for the generation of phylogenies in IQ-TREE[69] under LG + G4 model with 1,000 ultrafast bootstraps[68]. All alignments and phylogenies were manually inspected, eukaryotic organellar sequences were filtered out on the basis of the tree topologies and verified with BLASTP searches against the nr database, and seven widely distributed orthologous groups were selected for the final analysis. The final dataset contained the following eukaryotic (prokaryotic) orthologues: Ribosomal 40S S3 (Ribosomal 30S S3), Ribosomal 40S S15 (Ribosomal 30S S19), Ribosomal 60S L23 (Ribosomal 50S L14), Ribosomal 40S S11/14 (Ribosomal 30S S11), Ribosomal 40S S5/S7 (Ribosomal 30S S7), Ribosomal 40S S0/S2 (Ribosomal 30S S2) and RNA polymerase II subunit 3 (DNA-directed RNA polymerase subunit D). The trimmed alignments of the seven orthologues groups were concatenated with module Nexus of the Biopython package[75] to a final alignment of 1,097 amino acid sites, and was used to generate

the tree of life (Supplementary Data 7) with 1,000 ultrafast bootstrap[68] replicates in IQ-TREE[69] under LG + G4 model.

## Ancestral sequence prediction and ancestral aCR state estimation

Ancestral sequence reconstruction (ASR) was used to infer the most likely residues at the identified metal-preference determination positions ($X_{D-2}$ and $H/Q_{Cterm}$ or $Q_{Nterm}$) for the key ancestral nodes of SodFM1, SodFM2, SodFM3, SodFM4 and SodFM5 LCAs.

ASR was performed with the maximum likelihood phylogeny of 2,820 SodFMs (Supplementary Data 6) and the alignment used for the generation of the phylogeny (Supplementary Data 5). The reconstruction was performed using IQ-TREE[69] (-asr option) with the best fitting model used for the tree reconstruction (WAG + R10), and FastML[76] with WAG model. Both approaches gave similar results with only a second decimal place variation in posterior probability value between the two methods. Marginal and joint reconstruction in FastML predicted the same residues at the analysed positions of interest.

The possible ancestral aCR values for the key nodes of SodFM1 and SodFM2 ancestors were estimated using an IQ-TREE[69] maximum likelihood phylogeny of a representative sample of SodFMs with experimentally verified aCR values. The ancestral aCRs were estimated from the consensus phylogeny (of 1,000 ultrafast bootstrap[68] replicates) with a maximum likelihood approach implemented in fastAnc function of phytools package[77] and a Bayesian Markov chain Monte Carlo (MCMC) approach implemented in anc.Bayes[77] from the same package. To account for the uncertainty of the phylogenetic reconstruction, additional Bayesian estimation was performed in BayesTraits V4.0.0 (continuous random walk model A and MCMC analysis)[78,79] using 20,000 locally optimal trees (IQ-TREE -wt option)[69].

## Other bioinformatics methods

Mean sequence identities were calculated as an average of trimAL[67] pairwise identity comparisons (-sident) on MAFFT[66] alignments trimmed with trimAL[67] (gappyout mode). Sequence logos were generated with WebLogo3[80] and displayed so that the total height of each residue corresponds to the information content of the residue in the given position, and the relative size of multiple letters within each position corresponds to their relative frequency at this position. The phylogenetic trees were visualized and annotated with the protein presence/absence in the analysed genomes or with co-evolving amino acid residues using GraPhlAn[81]. Multiple sequence alignments were visualized and annotated using Jalview[82]. Phylogenetic trees were visualized and annotated using FigTree (http://tree.bio.ed.ac.uk/software/figtree/), Archaeopteryx[83].

## Reporting summary

Further information on research design is available in the Nature Portfolio Reporting Summary linked to this article.

## Data availability

Structural data that support the findings of this study have been deposited in the Protein Data Bank with the accession codes 8AVK, 8AVL, 8AVM, 8AVN (Supplementary Data 16). There are no restrictions on any data within the paper. Source data are provided with this paper.

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

## Acknowledgements

K.M.S., K.J.W. and T.E.K.-F. were supported by a grant from the National Institutes of Health (R01 AI155611) to T.E.K.-F. K.J.W. was also supported by a MAESTRO grant (2021/42/A/NZ1/00214) from the National Science Centre, Poland. A.B.-S. was supported by a Medical Research Council grant (MR/V032151/1). E.S.M. was supported by a PhD studentship from the Biotechnology and Biological Sciences Research Council. A.B. was funded by Newcastle University's Faculty of Medical Sciences. We thank Diamond Light Source for access to beamline I03 and I24 (mx18598). The contents of this work are solely the responsibilities of the authors and do not reflect the official views of any of the funders, who had no role in study design, data collection, analysis, decision to publish, or preparation of the manuscript.

## Author contributions

K.M.S., T.E.K.-F. and K.J.W. are the corresponding authors. K.M.S. and K.J.W. conceptualized and planned the study. K.M.S. and K.J.W. wrote the manuscript with editing by T.E.K.-F. and with contributions from all authors. K.M.S. performed all bioinformatic analyses. K.M.S. performed experimental work with contributions from A.B.-S. and E.S.M. A.B. crystallized and solved protein structures with contributions from A.B.-S. K.J.W. and T.E.K.-F. secured funding and managed the project.

## Competing interests

The authors declare no competing interests.

## Additional information

**Extended data** is available for this paper at https://doi.org/10.1038/s41559-023-02012-0.

**Correspondence and requests for materials** should be addressed to K. M. Sendra, T. E. Kehl-Fie or K. J. Waldron.

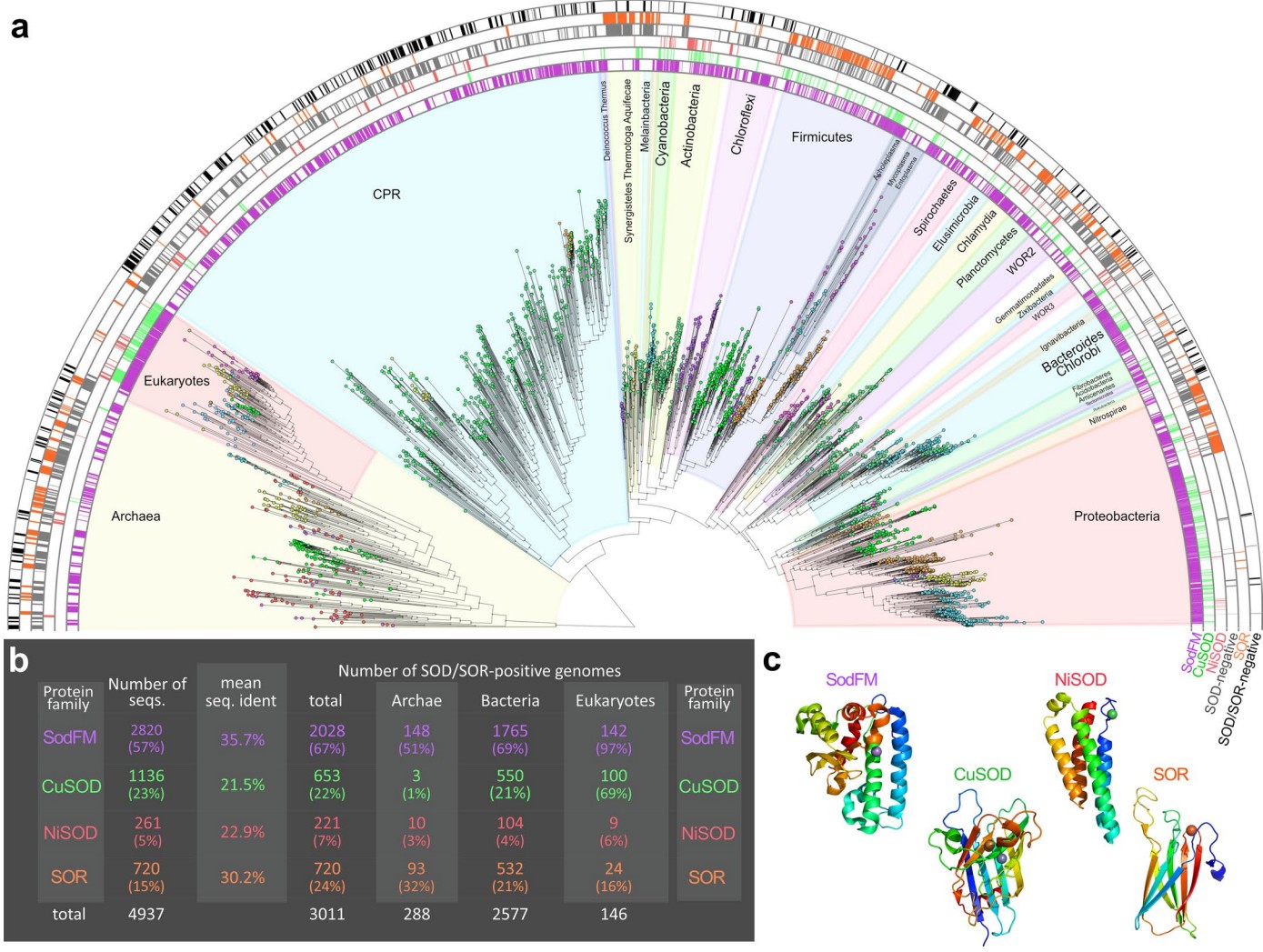

**Extended Data Fig. 1 | SodFMs constitute the largest and most widely distributed family of superoxide-detoxifying enzymes. a** The coloured semi-circles represent the distribution of Fe/Mn SODs (SodFMs; magenta), Cu/Zn SODs (green), and Ni SODs (red), mapped onto a phylogeny of 146 Eukaryotes, 2577 Bacteria, and 288 Archaea. SORs (superoxide reductases, orange) were often found in the genomes that lack any detectable SOD sequences (SOD-negative, grey; and Supplementary Data 17): in Archaea, including representatives from Euryarchaeota, Proteoarchaeota, TACK, and DPANN Archaea; Bacteria, with notable examples from classes Erysipelotrichia and Negativicutes within phylum Firmicutes; and Eukaryotic unicellular microbes *Spironucleus salmonicida* and *Giardia intestinalis*, well-studied examples of SOR-only eukaryotes[84–86]. Neither of the canonical superoxide-scavenging enzymes were detected in 389 out of 3011 analysed genomes (SOD/SOR-negative, black). This group included members of the phylum Tenericutes (highlighted in

grey within Firmicutes), such as the human pathogen *Mycoplasma pneumoniae*, which provides a known example of an aerobic organism without any identifiable SOD homologues encoded in its genome and without any measurable SOD activity in protein extracts[87,88]. **b** Table displaying the number of the identified protein sequences, their mean sequence identity, and the number of genomes encoding at least one representative of each of the protein families. Fe/Mn SODs constitute the most highly conserved (35.7% mean sequence identity), and the most common (found in 67.4% of all analysed genomes) family of superoxide-detoxifying enzymes found in all three domains of life (Eukaryotes, Bacteria, Archaea). **c** Each of the analysed protein families (Fe/Mn SOD, CuSOD, NiSOD, and SOR) utilises different metal cofactors and is structurally distinct, consistent with independent evolution of the capacity for superoxide detoxification within evolutionarily distinct protein folds.

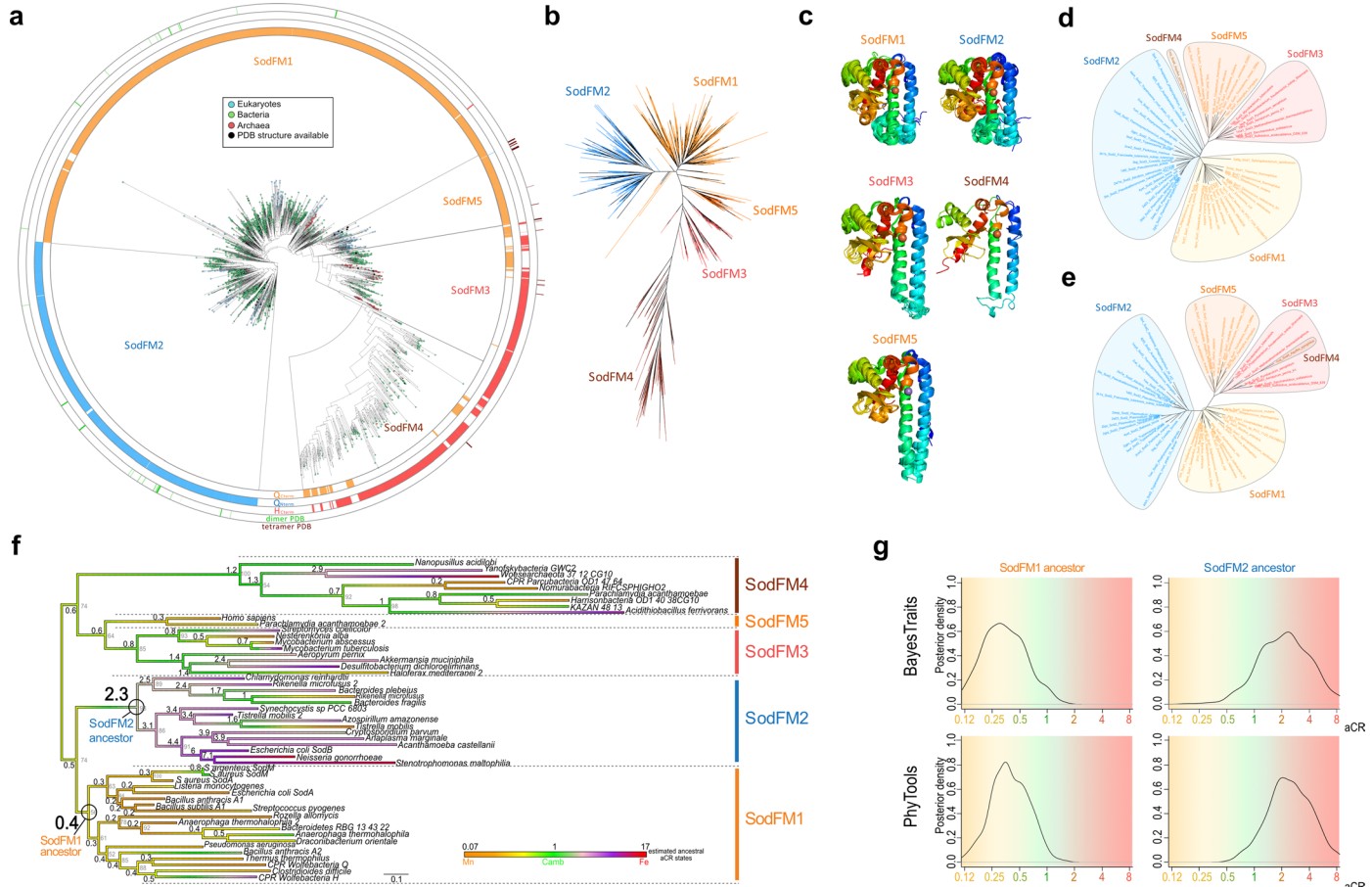

**Extended Data Fig. 2 | SodFM protein family can be subdivided into five distinct subfamilies based on phylogenetics, protein sequence, and 3D structure comparative analyses. a**. The rings, coloured to represent the presence of either of the three alternative water-coordinating residues, $H_{Cterm}$/$Q_{Cterm}$/$Q_{Nterm}$, within SOD catalytic centres (three innermost rings) and the oligomerisation state identified in available crystal structures (two outermost rings), were mapped onto the maximum likelihood protein tree of all SodFMs identified in our sampling of genomes from across the tree of life (Extended Data Fig. 1). N-terminal Gln ($Q_{Nterm}$: blue) is exclusively found in dimeric (green) SodFM2s, C-terminal water-coordinating Gln ($Q_{Cterm}$: orange) is found mainly in dimeric SodFM1s and tetrameric (brown) SodFM5s, and C-terminal His ($H_{Cterm}$: red) is most commonly (Fig. 2b) found in tetrameric (purple) SodFM3s and SodFM4s. **b**. An unrooted version of the SodFM protein tree presented in (a) **c**. Each cartoon represents a structural alignment of monomers of all SodFM1-5s with an available crystal structure in the protein data bank (PDB). The architecture of the two N-terminal α-helices (blue/turquoise/green) within monomers distinguishes the dimeric SodFM1s and SodFM2s from the tetrameric SodFM3s, SodFM4s, and SodFM5s. **d**. The tree represents a structural

comparison of SodFMs based on crystal structures available in the PDB, generated using a Distance matrix alignment (DALI). The DALI analysis reveals that SodFM subfamilies identified through phylogenetics (b) and amino acid sequence analysis (a) also form distinct groups when only their 3D structures are compared. **e**. Maximum likelihood phylogenetic protein tree generated using amino acid sequences of the SodFMs with available crystal structures used in the DALI analysis (d). **f**. The fastAnc estimates of the ancestral aCR states (black numbers, and tree colour coding) were mapped onto a consensus phylogeny of a representative sample of extant SodFMs with experimentally verified aCRs. Ultrafast bootstrap support values were presented for the key nodes (grey numbers) **g**. The frequency distributions of the ancestral SodFM1 (left panel) and SodFM2 (right panel) aCR state estimates from Bayesian analyses were presented as density plots on a log scale. Bayesian MCMC estimates from anc. Bayes function (PhyTools, bottom panel) were estimated from the consensus phylogeny (f). To account for the uncertainty of the phylogeny (f) BayesTraits analysis (top panel) was performed using 20,000 locally optimal trees generated with IQtree.

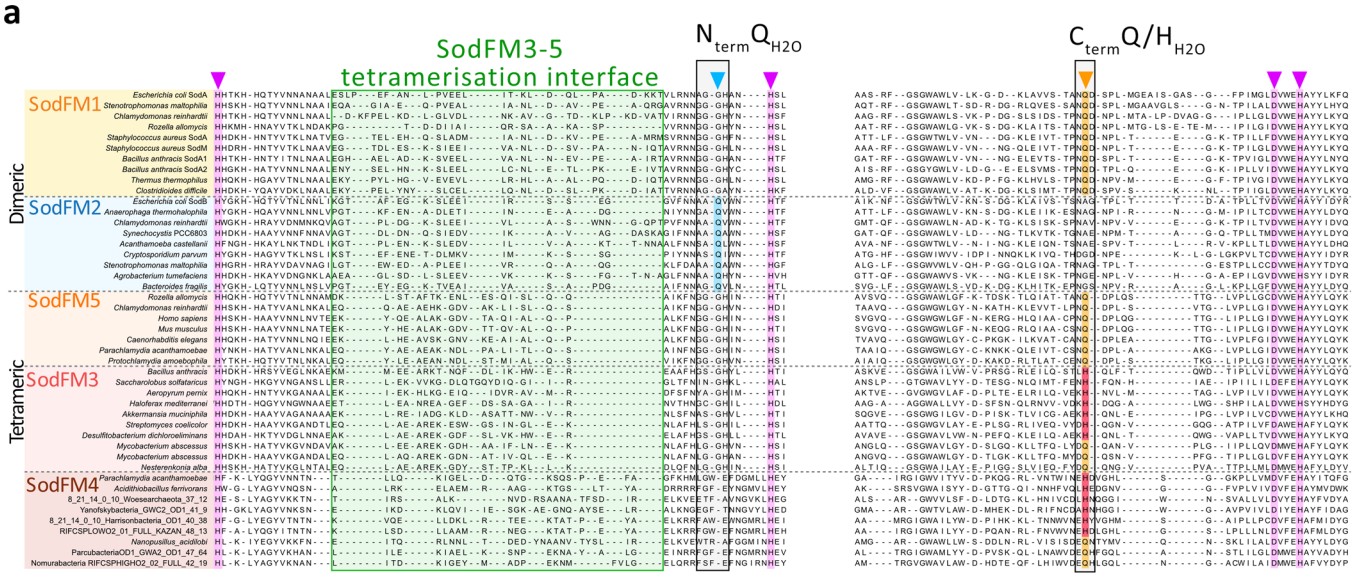

**Extended Data Fig. 3 | Position and identity of the conserved water-coordinating residue is not the sole determinant of SodFM1-5 phylogenetic grouping. a.** A key fragment of the multiple sequence alignment of representative amino acid sequences from all five SodFM subfamilies. The rectangles outline sequence regions that were removed in order to test the robustness of the SodFM1-5 groupings. This enabled confirmation that the distinct features of SodFM1-5s that are responsible for their grouping in protein phylogenies are encoded in sequence regions outside of the water coordinating Gln or His (Q/$H_{2O}$) and their immediate surroundings (black), and of the sequence region corresponding to the variable structural feature within the first two N-terminal α-helices (green) that distinguishes dimeric and tetrameric SodFMs (Extended Data Fig. 2a). Arrowheads indicate metal-coordinating

residues (turquoise), water-coordinating N-terminal Gln, $Q_{Nterm}$ (blue); C-terminal Gln, $Q_{Cterm}$ (orange); C-terminal His, $H_{Cterm}$ (red). **b.** The protein tree of SodFMs generated using a protein sequence alignment containing all regions of interest is the same as that used in previous figures (Extended Data Fig. 2a and b). **c-e.** Protein phylogenies generated after the removal of the regions containing the water-coordinating residue (c), the region encompassing the tetramerisation interface within two N-terminal helices (d), or all these regions at the same time (e) from the amino acid alignments (a) used to generate the trees. The overall SodFM protein tree topology has been retained in all trees (b-e), indicating that SodFM1-5 subfamily sequence-determinants are encoded within regions outside of the most apparent group-specific features.

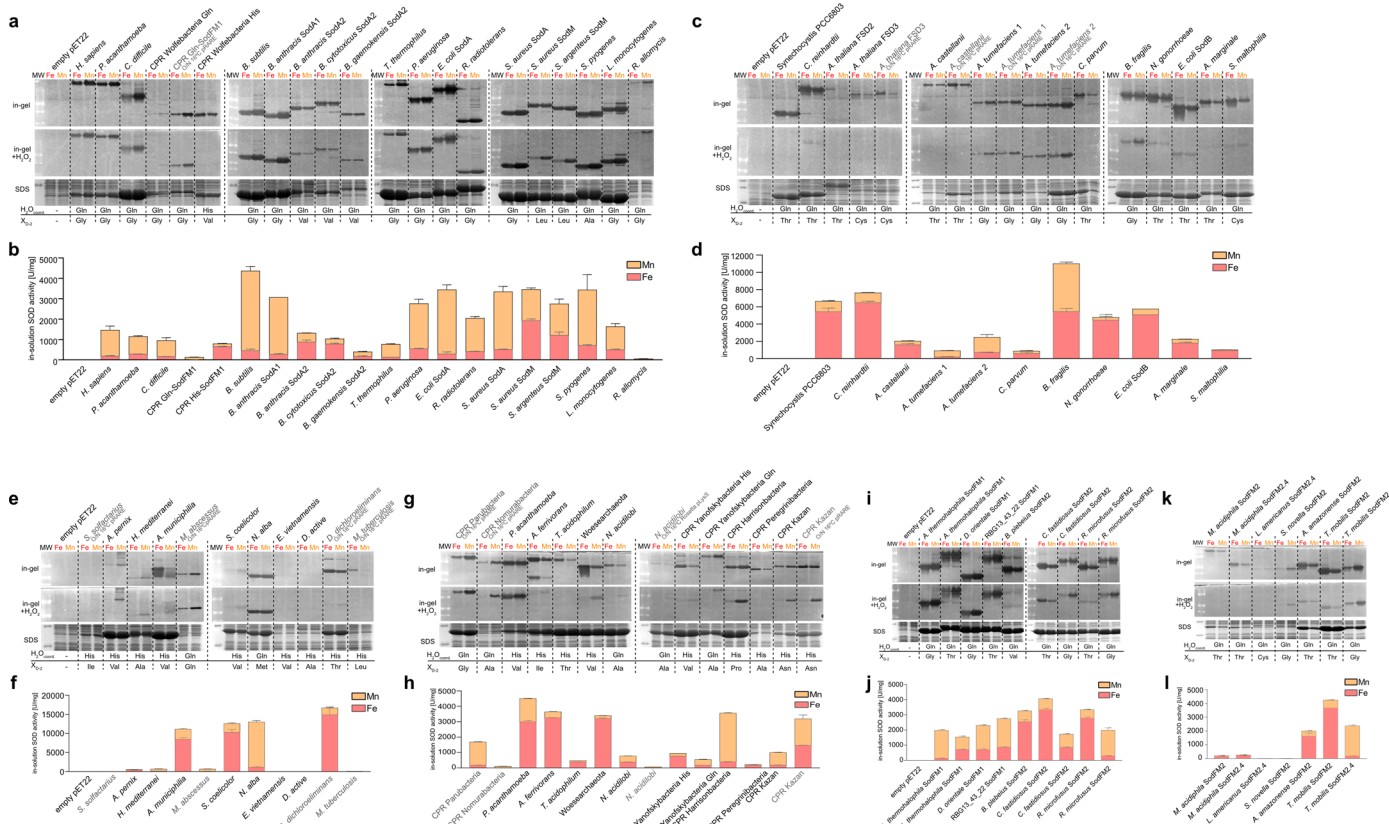

**Extended Data Fig. 4 | Investigating metal-preference and peroxide-sensitivity of natural SodFMs. a-h,** In-gel (a, c, e, g, i, k) and liquid (b, d, f, h, j, l) activity assay on soluble protein extracts from *E. coli* BL21(DE3) Δ*sodA*Δ*sodB* over-expressing SodFM1 and SodFM5s (a, b), SodFM2s (c,d), SodFM3s (e,f), SodFM4s (g,h). i-l. Experiments performed on expanded sampling of Bacteroidia SodFM1s and SodFM2s (i,j), and alphaproteobacterial SodFM2s (k,l), of interest identified in the bioinformatics analyses of metal-preference evolution. Proteins were expressed following 4 h IPTG induction in M9 medium supplemented with either 200 μM Fe (red) or 1 mM Mn (orange). Band patterns can be interpreted following the examples presented in the Extended Data Fig. 10. Negative controls consisted of the expression strain transformed with an empty vector (pET22). Some SodFMs (grey) were expressed at higher levels in *E. coli* BL21(DE3) Δ*sodA*Δ*sodB* transformed with pRARELysS plasmid over-night (O/N) at 16 °C. All in-gel activity assays (in-gel, and in-gel + $H_2O_2$) are native polyacrylamide

gels, and all Coomassie blue-stained protein gels (SDS) are denaturing gels. The identity of the water-coordinating residue ($H_2O_{coord}$) and that of the residue $X_{D-2}$ for each of the tested enzymes were annotated under the gels. Liquid SOD activity assay (b, d, f, h, j, l) was performed on the samples presented in (a, c, e, g, i, k). Bars represent average enzyme activities from assay performed in triplicate and are presented in a stacked format, therefore the total bar hight represents the sum of Fe- and Mn-activities, and the line separating the top bars (Mn activity, orange) from the bottom bars (Fe activity, red) can be interpreted as representing a relative proportion of the activity with each metal (cambialism ratio). Data are presented as mean values +/- standard deviations of three replicates. The presented values are expressed in U of SOD activity per mg of total protein measured using BCA assay. Full gels including molecular weight markers were included in Supplementary Data 10.

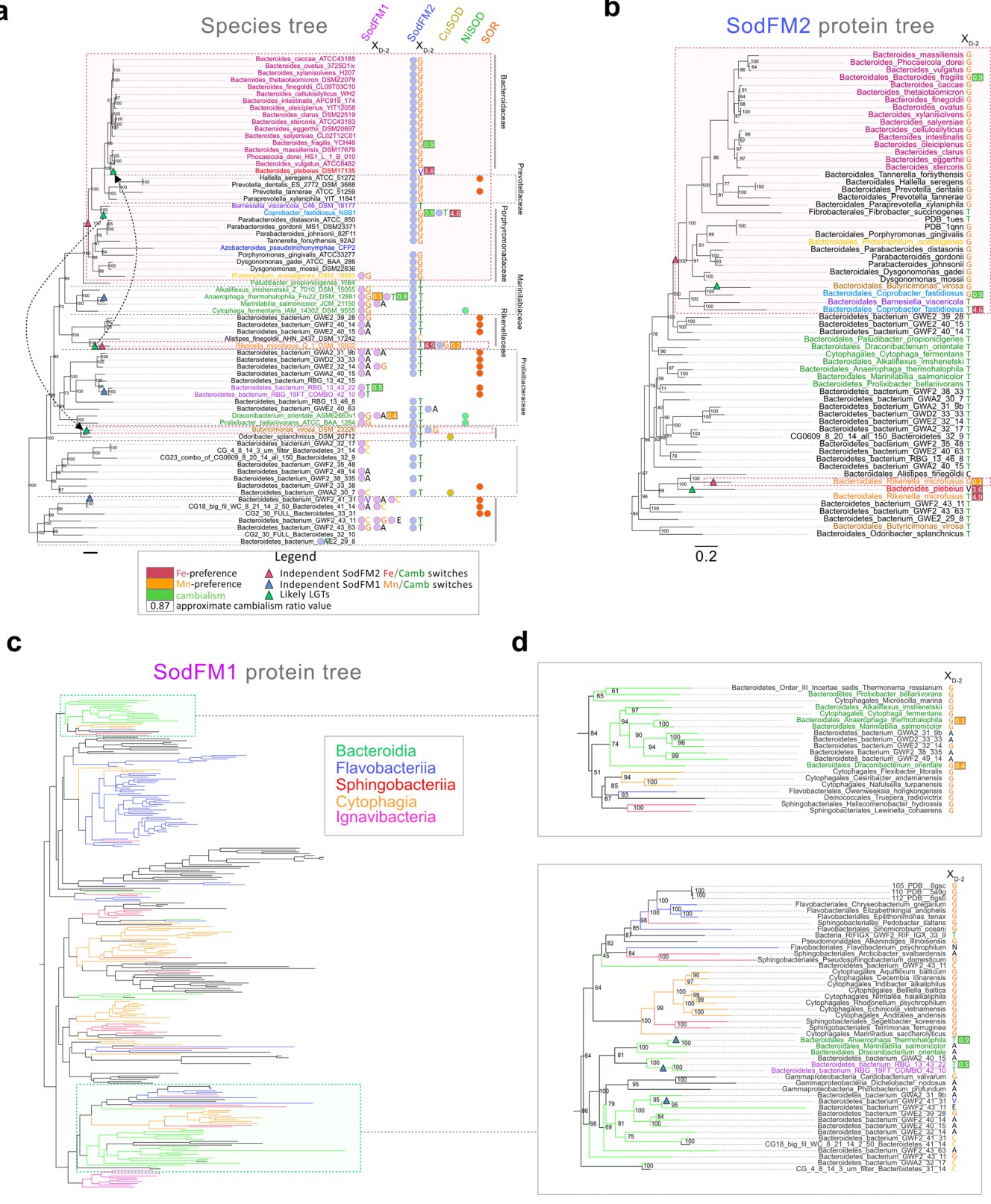

**Extended Data Fig. 5 | See next page for caption.**

**Extended Data Fig. 5 | Cambialism has evolved multiple independent times from two distinct SodFM subfamilies in Bacteroidales. a**. An enlarged version of the species tree of Bacteroidales (Fig. 3a) extracted from the tree of life (Fig. 1a). Colour-coded circles represent the presence of SodFM1 (pink), SodFM2 (blue), CuSOD (yellow), NiSOD (green), or SOR (orange). Coloured rectangles display experimentally verified aCRs (Fig. 2a, and Extended Data Fig. 4) for SodFM1 and SodFM2s. Red dashed line (a) outlines the lineages with SodFM2s containing a hallmark of higher Mn-preference, residue $G_{D-2}$. Coloured triangles indicate inferred ancestral Fe/Camb- (red), or Mn/Camb-switching (blue), and likely LGT (green) events. The likely source (dashed black line) and direction (arrowhead) of the LGTs were inferred based on the SodFM2 protein tree topology (b). Selected species names were colour coded to enable comparisons between the species tree (a) and the protein trees (b and d). **b-d**. Enlarged fragments of the tree of all SodFMs (Fig. 2a) containing Bacteroidia SodFM2s (b) and SodFM1s (c and d). Annotations correspond to those used in the species tree (A). Branches corresponding to SodFM1 sequences from classes of Bacteroidota and those from closely related Ignavibacteria were colour coded (c and d), whereas the black branches correspond to SodFM1s from other taxa. The regions of the SodFM1 tree containing Bacteroidia sequences of interest were outlined with green dashed lines (c), and their magnified versions were presented in (d).

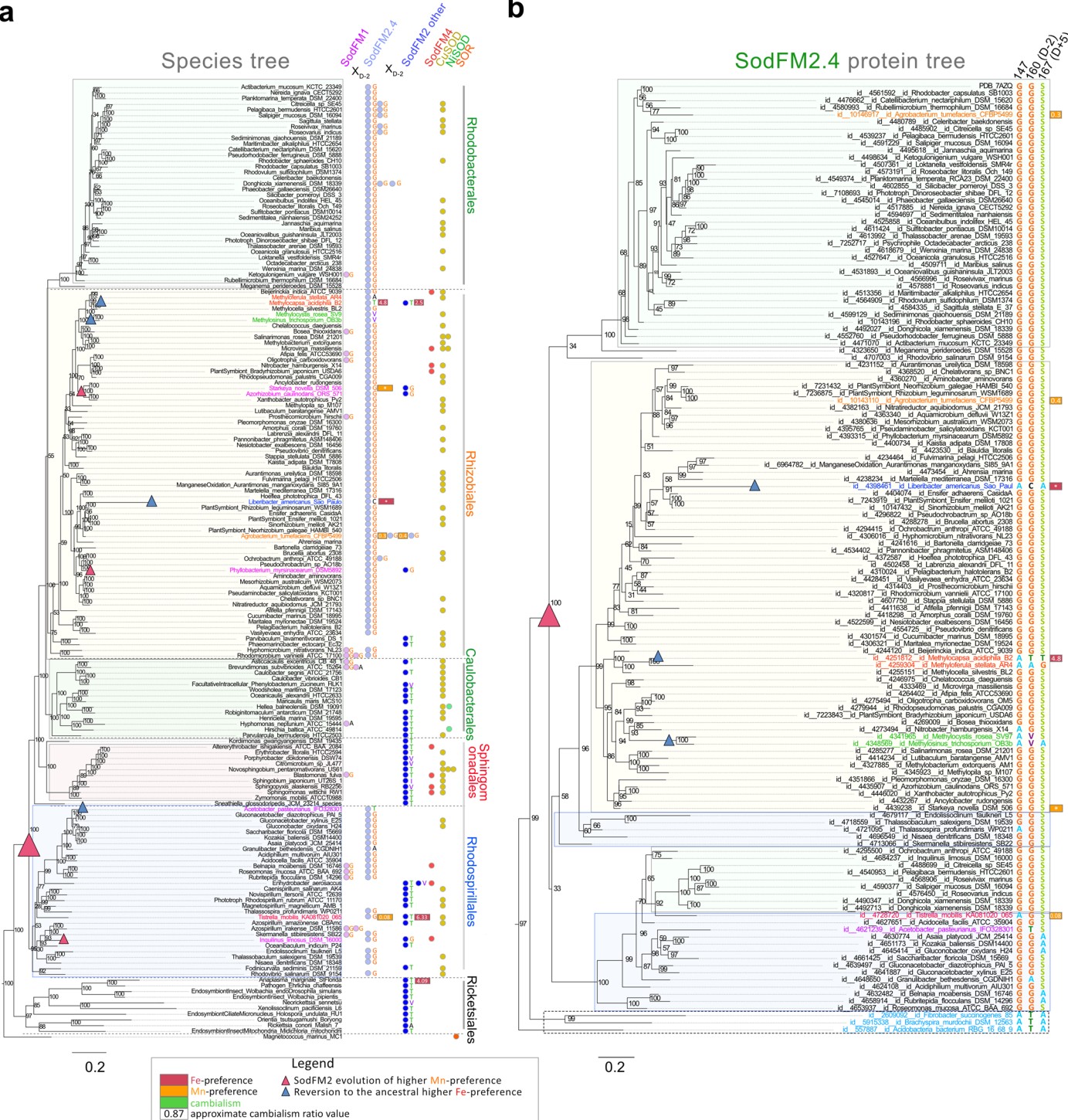

**Extended Data Fig. 6 | Alphaproteobacteria-specific SodFM2.4 was acquired in their last common ancestor. a**. An enlarged fragment of the species tree of Alphaproteobacteria (Fig. 3b) extracted from the tree of life (Fig. 1a). Colour-coded circles represent the presence of SodFM1 (pink), SodFM2.4 (light blue), a SodFM2 other than SodFM2.4 (dark blue), SodFM4 (red), CuSOD (yellow), NiSOD (green), or SOR (orange). Coloured rectangles display experimentally verified aCRs for SodFM1 and SodFM2s (Fig. 2a, and Extended Data Fig. 8). Coloured triangles indicate inferred acquisitions of higher Mn-preference (red), or reversions back to the most likely ancestral Fe-preference (blue). The likely acquisition of the ancestral SodFM2.4 in the last common ancestor of the Alphaproteobacteria following the split from Ricketsiales is indicated (large pink triangle). Selected species names were colour-coded to enable comparisons

between the species tree (a) and the protein tree (b). **b**. Zoomed in fragments of the tree of all SodFMs (Fig. 2a) containing all SodFM2.4s. Annotations correspond to those used in the species tree (a). The letters mapped onto the protein tree (positions 147, 160, and 167; numbering as described in Extended Data Fig. 8) correspond to the SodFM2.4-specific residues identified in the amino acid correlation analysis (Extended Data Fig. 8a). The overall topology of the protein tree (b) resembles that of the species tree (a), consistent with ancient acquisition of a SodFM2.4 (large pink triangle, and Fig. 3F) with an increased Mn-preference followed by vertical inheritance. Grouping of the Fe-preferring SodFM2.4s (blue triangles) alongside the closely related canonical SodFM2.4s provides further support for the hypothesis that the Fe-preferring SodFM2.4s represent evolutionary reversion to their ancestral Fe-preference.

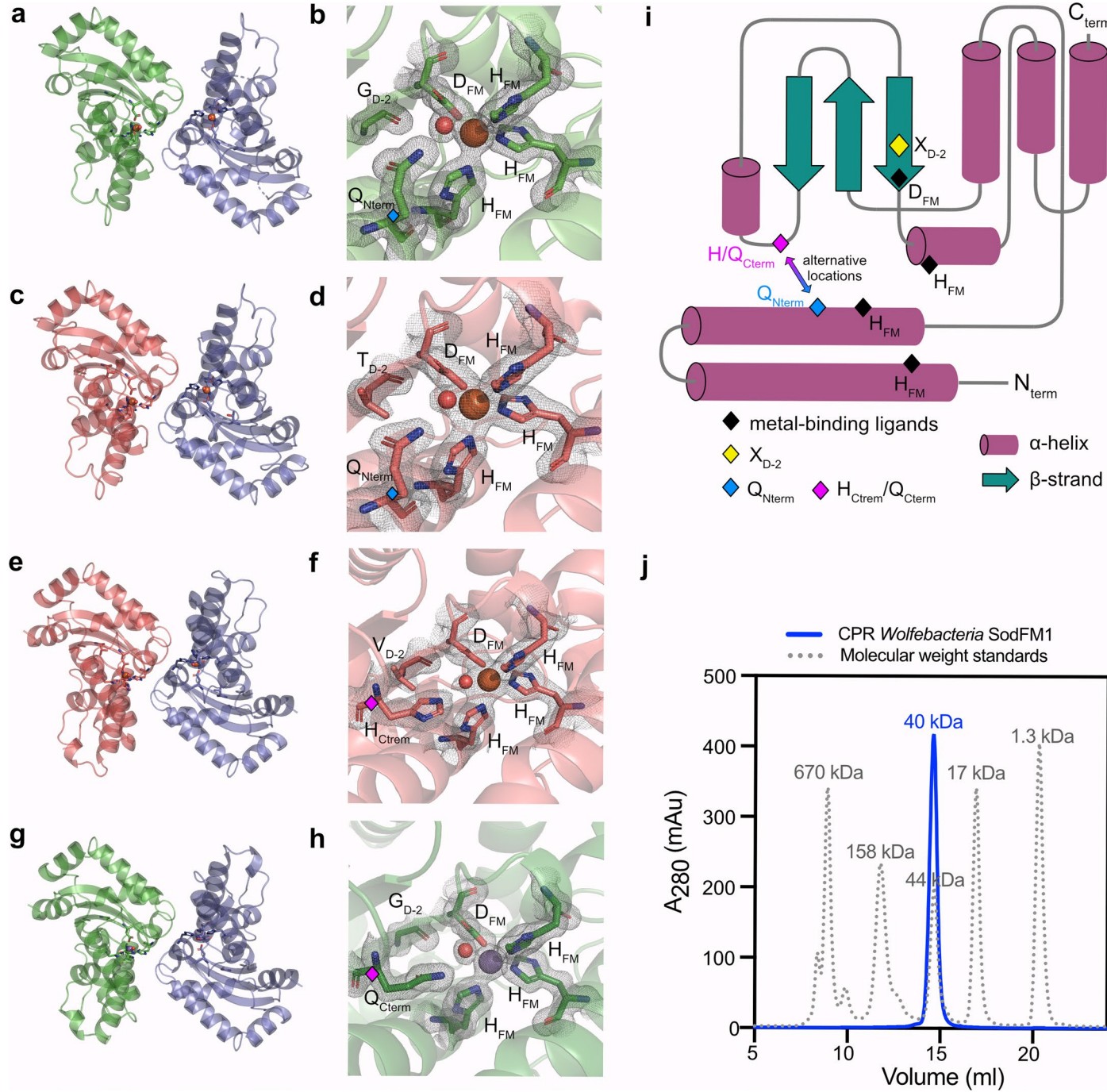

**Extended Data Fig. 7 | Structural investigation of SodFM mutants with changed metal-preference. a-h.** Overall architecture (a, c, e, g) and enlarged view of the catalytic centre (b, d, f, h) of *B. fragilis* wild type (WT) camb-SodFM2 (a, b), its Fe-$G_{D-2}$T,$F_{D-1}$C mutant (c, d), and CPR Wolfebacteria Fe-SodFM1 (e, f) and its camb-H/Q,$V_{D-2}$G mutant (g, h). The grey mesh surrounding stick representations of residues of interest correspond to 2Fo-Fc electron density map in the solved structures. Spheres represent iron (orange), manganese (blue), and water (red) molecules. *B. fragilis* structure was solved for double $G_{D-2}$T,$F_{D-1}$C mutant for consistency with previously solved structures of *S. aureus* SodFM mutants (PDB 6qv8, and 6qv9). The mutation $F_{D-1}$C had no apparent effect on metal-preference compared to that of the *B. fragilis* $G_{D-2}$T single

mutant (Extended Data Fig. 9c,d). **i.** General secondary structure topology diagram of SodFMs displaying the position of the key residues involved in: metal coordination (H/$D_{FM}$, black), metal-preference determination $X_{D-2}$ (yellow), and two alternative locations of catalytic water coordinating H/Q within either C-terminal ($H_{Cterm}$/$Q_{Cterm}$, magenta) or N-terminal ($Q_{Nterm}$, blue) domains. Cylinders represent α-helices and arrows represent β-strands. **j.** Analytical size-exclusion chromatography of WT CPR Wolfebacteria Fe-SodFM1. The protein elution pattern (j) is consistent with the homodimeric architecture observed in its crystal structure (e) despite containing water-coordinating His residue common in homotetrameric SodFM3s/SodFM4s.

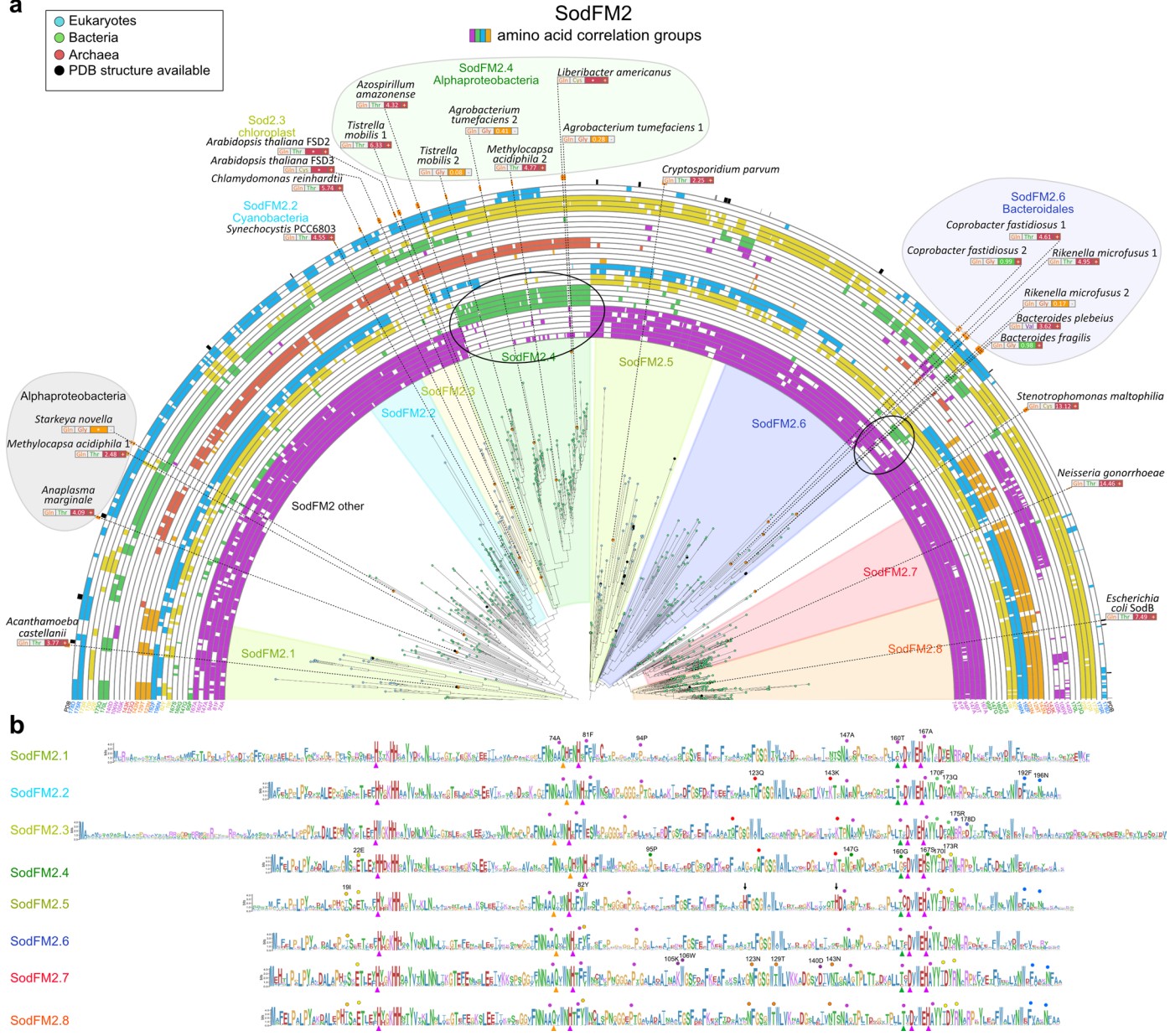

**Extended Data Fig. 8 | Signatures of metal-preference switching can be detected in amino acid correlation analysis of SodFM2 subfamily members.**
**a.** Residues identified in amino acid correlation analysis of SodFM2s were colour coded and mapped onto the SodFM2 protein phylogeny as separate semi-circles. Multiple amino acid correlation groups with distinct distribution patterns across the SodFM2 tree were identified. The alphaproteobacterial SodFM2.4s constituted the most distinct subgroup of SodFM2s, which grouped closely with eukaryotic chloroplast SodFM2.3s and Cyanobacterial SodFM2.2s. Alongside Bacteroidales SodFM2.6, SodFM2.4s also represented one of the two groups with strong patterns of evolutionary metal-preference switching (black elliptical outlines) identified in our dataset. Characteristically in these two groups, residues from the first correlation group (innermost magenta semi-circles) were often replaced with those from the second group (the innermost green semi-

circle), one of which was the key metal-preference determinant $X_{D-2}$ (Figs. 1c and 2c). The top ring (black) indicates SODs with available crystal structures. Species names and the outermost semi-circle (orange) indicated the SODs with aCR values experimentally verified in this study (Fig. 2a). **b.** Sequence logos of each SodFM2 sub-group, with amino acid correlation groups mapped onto the logos as coloured semi-circles. Within the analysed SodFM2 alignment residue $X_{D-2}$ is at position 160. Black arrows indicated two highly distinct His residues identified in SodFM2.5, which contained many sequences from diverse Eukaryotic microbial pathogens including *Plasmodium falciparum, Toxoplasma gondii, Trypanosoma brucei*, and *Cryptosporidium parvum*. Arrowheads under each logo signify metal coordinating residues (magenta), catalytic water coordinating $Q_{Nterm}$ (orange), and the residue $X_{D-2}$ (green).

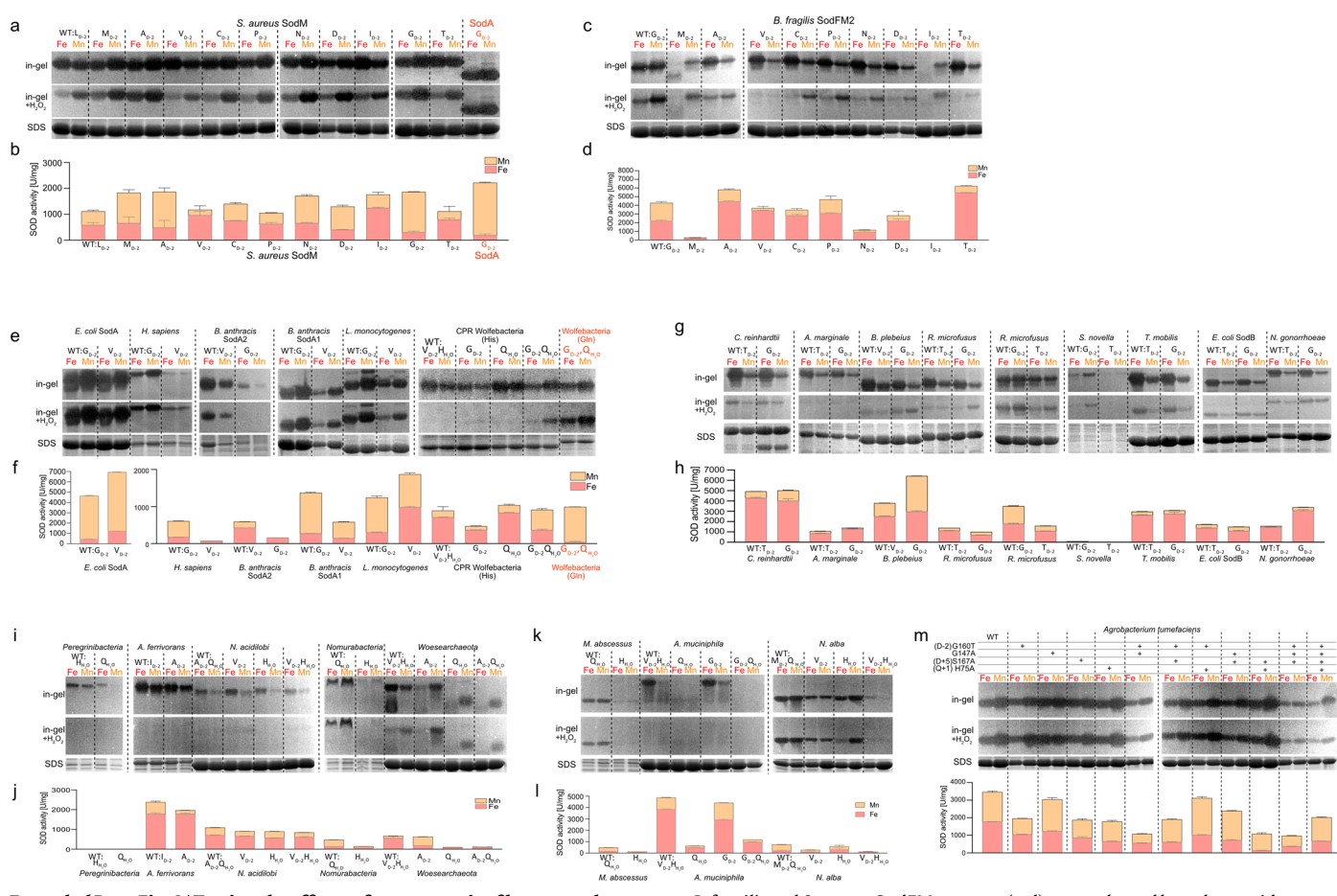

**Extended Data Fig. 9 | Testing the effects of mutagenesis of key secondary coordination sphere residues on metal-preference of SodFMs. a-l,** In-gel (a, c, e, g, i, k) and liquid (b, d, f, h, j, l) activity assay on soluble protein extracts from *E. coli* BL21(DE3) Δ*sodA*Δ*sodB* over-expressing SodFM mutants and wild type (WT) controls. **a,b** *S. aureus* cambialistic SodFM1 (SodM), and its $X_{D-2}$ mutants, and canonical *S. aureus* Mn-SodFM1 (SodA) representing likely ancestral state of the SodA/SodM last common ancestor. **c,d** *B. fragilis* cambialistic SodFM2, and its $X_{D-2}$ mutants. The $X_{D-2}$ residues substituted in *B. fragilis* and *S. aureus* SodFM mutants (a,d) were selected based on residues found in other natural enzymes at this position (Fig. 2b). **e-l,** Mutants of: SodFM1s and *Homo sapiens* SodFM5 (e,f), SodFM2s (g,h), SodFM4s (i,j), SodFM3s (k,l). **m,** Mutagenesis of SodFM2.4-specific residues (Extended Data Fig. 8) in *A. tumefaciens* Mn-SodFM2.4. The results were analysed and presented as described in Extended Data Fig. 4. Identity of the mutated residues were indicated above the gels. Full gels including molecular weight markers were included in Supplementary File 10.

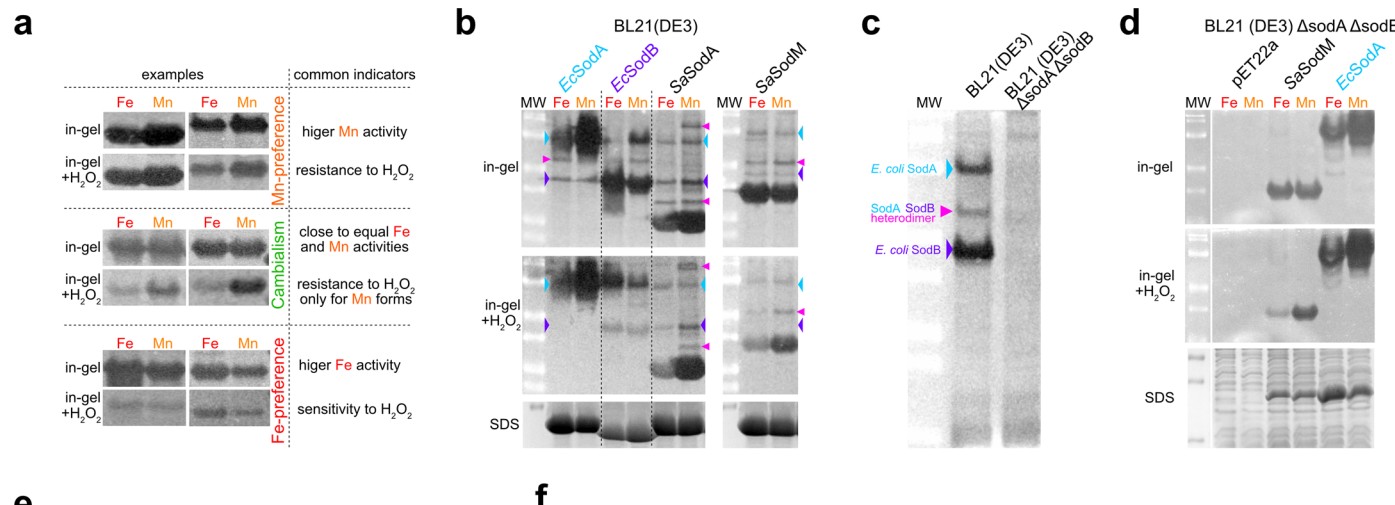

| SodFM | metal | activity | liquid aCR (in-gel aCR) | metal-verified protein preparations | | | | | | |
|---|---|---|---|---|---|---|---|---|---|---|
| | | | | this study | | | literature data | | | |
| | | | | activity | metal-loading | | CR | activity | metal-loading | | CR |
| *S. aureus* SodA | Fe | 15% ± 1% | 0.18 (0.03) | 8% ± 0% | 1% Mn | 73% Fe | 0.09 | 0.2% | 0% Mn | 100% Fe | 0.002 |
| | Mn | 85% ± 8% | | 92% ± 1% | 98% Mn | 2% Fe | | 99.8% | 100% Mn | 0% Fe | |
| *S. aureus* SodM | Fe | 56% ± 2% | 1.26 (1.15) | 59% ± 3% | 0.5% Mn | 100% Fe | 1.41 | 50% | 0% Mn | 100% Fe | 1.00 |
| | Mn | 44% ± 2% | | 41% ± 9% | 100% Mn | 0.7% Fe | | 50% | 100% Mn | 0% Fe | |
| *E. coli* SodA | Fe | 9% ± 3% | 0.09 (0.01) | 0% ± 0% | 0% Mn | 2% Fe | 0.003 | 0% | 0% Mn | 95% Fe | 0.000 |
| | Mn | 91% ± 7% | | 100% ± 3% | 98% Mn | 2% Fe | | 100% | 98% Mn | 0% Fe | |
| *E. coli* SodB | Fe | 88% ± 0% | 7.49 (56.9) | 94% ± 9% | 0.4% Mn | 84% Fe | 15.8 | 99.9% | 0% Mn | 98% Fe | 837.5 |
| | Mn | 12% ± 0% | | 6% ± 1% | 89% Mn | 2% Fe | | 0.1% | 50-98% Mn | 0% Fe | |
| *B. fragilis* | Fe | 50% ± 3% | 0.98 (1.34) | 42% ± 0% | 3% Mn | 93% Fe | 0.73 | 59% | ? % Mn | 90% Fe | 1.42 |
| | Mn | 50% ± 2% | | 58% ± 0% | 90% Mn | 10% Fe | | 41% | 50% Mn | ?% Fe | |

**Extended Data Fig. 10 | Combination of a standardised SOD activity assay with SodFM-free protein expression strain provides a reliable method for metal-preference determination. a**. Examples of in-gel activity assay results (in-gel) for Fe-, Mn-, and cambialistic-SodFMs with common indicators of each metal-preference and resistance to peroxide (H₂O₂). **b**. In-gel SOD activity assay on total protein extracts from wild-type (WT) *E. coli* BL21(DE3) overexpressing SodFMs(*Ec*SodA and *Ec*SodB, and *Sa*SodA and *Sa*SodM) in the presence of Fe (red) or Mn (orange). Samples contain endogenous *Ec*SodA (turquoise arrowheads) and *Ec*SodB (purple) expressed from the WT genome, as well as numerous heterodimeric forms (magenta) assembled from monomers of *Ec*SodA, *Ec*SodB, and the overexpressed enzymes. **c,d**. To address the issue of the contaminating endogenous *E. coli* SodFMs, we generated *E. coli* BL21(DE3) Δ*sodA*Δ*sodB* clean double knock out strain (ΔΔ*sodAB*). No endogenous SodFM activity bands were observed in total protein extracts from ΔΔ*sodAB* (c, right lane) and in those from ΔΔ*sodAB* overexpressing SodFM isozymes (**d**, *E. coli* SodA and *S. aureus* SodM) in the presence of Fe (red) or Mn (orange), and a negative control transformed with empty plasmid (pET22a). **e**. In-gel activity assay on soluble protein extracts from SodFM (SaSodA, and SaSodM) expressing ΔΔ*sodAB* cultured in the presence of varying metal concentrations added to the M9 medium indicates that metal supplementation is required to metal-load overexpressed SodFMs. Experiments 7a-7d constituted routine controls and were reproduced more than three times, experiment 7e was independently reproduced two times. **f**. Comparison of aCR values estimated from the liquid SOD assay on soluble protein extract (liquid aCR), aCR from in-gel band intensities using the same samples (in-gel aCR), and 'true' CR samples calculated using metal-verified protein preparations generated here (this study) or from the previously published data (literature data, where '?' represents unreported values).

# Reporting Summary

## Statistics

For all statistical analyses, confirm that the following items are present in the figure legend, table legend, main text, or Methods section.

| n/a | Confirmed | |
|---|---|---|
| ☐ | ☒ | The exact sample size (*n*) for each experimental group/condition, given as a discrete number and unit of measurement |
| ☐ | ☒ | A statement on whether measurements were taken from distinct samples or whether the same sample was measured repeatedly |
| ☐ | ☒ | The statistical test(s) used AND whether they are one- or two-sided *Only common tests should be described solely by name; describe more complex techniques in the Methods section.* |
| ☒ | ☐ | A description of all covariates tested |
| ☒ | ☐ | A description of any assumptions or corrections, such as tests of normality and adjustment for multiple comparisons |
| ☒ | ☐ | A full description of the statistical parameters including central tendency (e.g. means) or other basic estimates (e.g. regression coefficient) AND variation (e.g. standard deviation) or associated estimates of uncertainty (e.g. confidence intervals) |
| ☒ | ☐ | For null hypothesis testing, the test statistic (e.g. *F*, *t*, *r*) with confidence intervals, effect sizes, degrees of freedom and *P* value noted *Give P values as exact values whenever suitable.* |
| ☒ | ☐ | For Bayesian analysis, information on the choice of priors and Markov chain Monte Carlo settings |
| ☒ | ☐ | For hierarchical and complex designs, identification of the appropriate level for tests and full reporting of outcomes |
| ☒ | ☐ | Estimates of effect sizes (e.g. Cohen's *d*, Pearson's *r*), indicating how they were calculated |

*Our web collection on statistics for biologists contains articles on many of the points above.*

## Software and code

Policy information about availability of computer code

| Data collection | Not applicable. |
|---|---|
| Data analysis | Bioinformatic analyses used common commercial or publicly available databases/code/programs: PDB, NCBI, Pfam, BlastP, MAFFT, trimAL, IQ-TREE, ModelFinder, PyMol, DaliLiteV5.1, PFstats, HMMER3.3,hmmsearch, module Nexus of the Biopython package, WebLogo3, GraPhlAn, Jalview, FigTree and Archaeopteryx. |

For manuscripts utilizing custom algorithms or software that are central to the research but not yet described in published literature, software must be made available to editors and reviewers. We strongly encourage code deposition in a community repository (e.g. GitHub). See the Nature Portfolio guidelines for submitting code & software for further information.

## Data

Policy information about availability of data

All manuscripts must include a data availability statement. This statement should provide the following information, where applicable:
- Accession codes, unique identifiers, or web links for publicly available datasets
- A description of any restrictions on data availability
- For clinical datasets or third party data, please ensure that the statement adheres to our policy

Structural data that support the findings of this study have been deposited in the Protein Data Bank with the accession codes 8AVK, 8AVL, 8AVM, 8AVN (see Source Data Table 6). Source data for are provided as Source Data Files and Tables with the paper. There are no restrictions on any data within the manuscript.

## Human research participants

Policy information about studies involving human research participants and Sex and Gender in Research.

| | |
|---|---|
| Reporting on sex and gender | Not applicable to this study. |
| Population characteristics | *Describe the covariate-relevant population characteristics of the human research participants (e.g. age, genotypic information, past and current diagnosis and treatment categories). If you filled out the behavioural & social sciences study design questions and have nothing to add here, write "See above."* |
| Recruitment | *Describe how participants were recruited. Outline any potential self-selection bias or other biases that may be present and how these are likely to impact results.* |
| Ethics oversight | *Identify the organization(s) that approved the study protocol.* |

Note that full information on the approval of the study protocol must also be provided in the manuscript.

# Field-specific reporting

Please select the one below that is the best fit for your research. If you are not sure, read the appropriate sections before making your selection.

☒ Life sciences      ☐ Behavioural & social sciences      ☐ Ecological, evolutionary & environmental sciences

For a reference copy of the document with all sections, see nature.com/documents/nr-reporting-summary-flat.pdf

# Life sciences study design

All studies must disclose on these points even when the disclosure is negative.

| | |
|---|---|
| Sample size | Biochemical experiments were performed n=3 to enable  error determination, consistent with the standard for these types of experiments. |
| Data exclusions | No data were excluded. |
| Replication | All biochemical experiments were replicated n=3 and were found to be reproducible. |
| Randomization | No randomisation was applied as it was no suitable for these biochemical experiments. |
| Blinding | No blinding was applied as it was not suitable for these biochemical experiments. |

# Reporting for specific materials, systems and methods

We require information from authors about some types of materials, experimental systems and methods used in many studies. Here, indicate whether each material, system or method listed is relevant to your study. If you are not sure if a list item applies to your research, read the appropriate section before selecting a response.

### Materials & experimental systems

| n/a | Involved in the study |
|---|---|
| ☒ ☐ | Antibodies |
| ☒ ☐ | Eukaryotic cell lines |
| ☒ ☐ | Palaeontology and archaeology |
| ☒ ☐ | Animals and other organisms |
| ☒ ☐ | Clinical data |
| ☒ ☐ | Dual use research of concern |

### Methods

| n/a | Involved in the study |
|---|---|
| ☒ ☐ | ChIP-seq |
| ☒ ☐ | Flow cytometry |
| ☒ ☐ | MRI-based neuroimaging |

