## [Peer Review File · Nature Ecology & Evolution]

Peer Review Information

Journal: Nature Ecology & Evolution

Manuscript Title: An ancient metalloenzyme evolves through metal preference modulation

Corresponding author name(s): An ancient metalloenzyme evolves through metal preference modulation

Editorial Notes:

Reviewer Comments & Decisions:

Decision Letter, initial version:

30th November 2022

Dear Kevin,

Your manuscript entitled "An ancient metalloenzyme evolves through metal preference modulation" has now been seen by 3 reviewers, whose comments are attached. The reviewers have raised a number of concerns which will need to be addressed before we can offer publication in Nature Ecology & Evolution. We will therefore need to see your responses to the criticisms raised and to some editorial concerns, along with a revised manuscript, before we can reach a final decision regarding publication.

We therefore invite you to revise your manuscript taking into account all reviewer and editor comments. Please highlight all changes in the manuscript text file in Microsoft Word format.

* If you have not done so already please begin to revise your manuscript so that it conforms to our Article format instructions at <http://www.nature.com/natecolevol/info/final-submission>. Refer also to any guidelines provided in this letter.

2[REDACTED]

Nature Ecology & Evolution is committed to improving transparency in authorship. As part of our efforts in this direction, we are now requesting that all authors identified as 'corresponding author' on published papers create and link their Open Researcher and Contributor Identifier (ORCID) with their account on the Manuscript Tracking System (MTS), prior to acceptance. ORCID helps the scientific community achieve unambiguous attribution of all scholarly contributions. You can create and link your ORCID from the home page of the MTS by clicking on 'Modify my Springer Nature account'. For more information please visit www.springernature.com/orcid.

[REDACTED]

Reviewer expertise:

Reviewer #1: metalloproteins, computational, structural biology

Reviewer #2: protein evolution, phylogenetics, experimental and computational

Reviewer #3: biochemistry and structure of metalloproteins

Reviewers' comments:

Reviewer #1 (Remarks to the Author):

In this paper, the authors use a combination of bioinformatics and biochemistry methods to show how the metal preference of Fe/Mn SOD can be finely tuned, and how this actually happened multiple times across evolution. The methods are appropriate, the results are convincing and discussed clearly. Therefore, I recommend publication in Nature Ecology & Evolution.

2Reviewer #2 (Remarks to the Author):

Sendra and colleagues present a very interesting study on the evolution of metal specificity in superoxide dismutases. They employ phylogenetics and extensive biochemistry to show that metal preferences frequently switch and provide some mechanistic insights about how these switches could happen. This is one of very few studies that investigate a biochemical phenotype across an entire phylogeny – an approach that I hope will become much more common in the future. The conclusions are interesting and well supported by the data. I believe this work will be broadly interesting to evolutionary biologists and biochemists and as such is highly suitable for the broad readership of Nature Ecology and Evolution.

The manuscript contains extensive amounts of data and is difficult to read at times. One problem is that the paper frequently jumps between different Extended Data figures and that it does not go through them in order. A less important problem is that there was no merged document that contained all figures (including Extended Data) with their corresponding legends beneath them. This reviewer found the switching between documents and scrolling very exhausting while reviewing the work.

I have a few minor comments:

- 1) What was the model selection criterion in IQ-tree. I assume it was BIC. This should be described
- 2) The trees in the main text figures 1 and 2 lack a scale bar.
- 3) I would urge the authors to switch from a semi circle representation of the trees to a normal 'left to right' representation. Semi circles are incredibly hard to read. I see no advantage of this representation and many disadvantages (including having to tilt my head to read some of the labels and struggling to grasp the relationships between major groups on the tree).
- 4) The cambialism score should be a log score. The authors already plot it on log scales in various figures. It is much easier to compare relative preferences between Fe and Mn on a log scale
- 5) Figure 2C overloaded me with different colors and I had to stare at it for a long time to understand what it is supposed to convey. Perhaps this could be simplified.
- 6) The authors quantify differences in activity to make statements about metal preference. Are there perhaps also differences in metal affinity and do they always correlate with differences in activity? There is an assumption in the paper that the observed preferences are somehow optimal for each environment. I'm not sure that statement can be made without knowing the environmental abundances of these metals
- 7) The authors make various statements about the evolution of different metal specificities and ancestral specificities, but never use their trees to explicitly reconstruct the evolution of this trait. I think the authors should consider performing a trait reconstruction using their data (i.e. use parsimony or a gaussian model to infer the aCR ratio at internal nodes of their protein tree). The paper would live without it, but since this is a 'first of its kind' study, it would be great to set a good precedent.
- 8) The paper mentions a correlation between different amino acids and the aCR ratio. This language is somewhat unfortunate. To make this statement, a phylogenetic regression would have to be performed. A straight correlation ignores the relatedness of the sequences of the tree and can lead to erroneous inferences of correlations. If the authors do not want to perform a phylogenetic regression, they should modify this language.

39) The paper uses the term 'evolutionary hotspot' in a manner that I found confusing. I think the authors mean taxonomic groups in which preferences frequently switch. But this term has previously been used to describe regions of proteins where important changes tend to happen. In order to avoid confusion, perhaps the use of this term could be reconsidered.

10) In line 278 there is a sentence about selection for cambialism. I could not understand how the data support this statement. Either it needs clarification or be removed.

11) I found the last few sentences of the discussion a bit over the top. I don't understand how these proteins have anything to do with global warming or epidemics. The work presented here is a major advance in the field of evolutionary biochemistry. I don't think these grandiose statements are necessary to sell it.

Reviewer #3 (Remarks to the Author):

In this manuscript, Dendra et al. present a comprehensive evolutionary analysis of metal preference of the SodFM superfamily of superoxide dismutases that span that entire tree of life. This work derives from previous work by the same investigator team (refs 4-5) and their discovery and characterization of a truly cambialistic SodM from *Staphylococcus aureus*, which is capable of using Fe or Mn as the metal cofactor for superoxide anion dismutation. That prior work lead to the question posed here: What are the evolutionary mechanisms and timescales by which Fe- vs Mn- vs. cambialistic SODs evolved? This is a broadly important question in metalloenzyme catalysis and evolution and the SodFM superfamily is one of a handful of outstanding systems with which to explore this at level of detail required to obtain these insights.

The conclusions that the authors reach appear to be well-supported by the data, and of high interest to those interested in metalloproteomes and metalloenzymology alike. The overall approach is to construct a suitable tree and then purify many (≥ 65) enzymes from each of five main subfamilies (SodFM1-5), or within a single group, and measure the aCR, or approximate cambialism ratio, of recombinant SODs purified as Mn or Fe-laded enzymes.

A couple of key takeaways are well-supported by the data. Phylogenetic grouping on the basis global sequence analysis cannot be used to assess metal-preference. Further, although earliest common ancestors identified here may have been cambialistic or characterized by a particular metal preference, shifts in metal preferences typically occur multiple times during the course of evolution within a main SodFM family. The authors describe this a "sliding scale" of adaptability rather than clearly defined evolutionary path, an important finding.

The authors go on to identify candidate structural features that impact metal specificity, focusing on a few key determinants (XD-2; HCterm/QCterm/QNterm). They provide compelling support for the idea that these residue play important (albeit, not exclusive) roles in metal-preference switching. The importance of the identity of X-2 residue, in particular, was tested biochemically and the model validated. Changes here can push metal preference in one direction exclusively or another or both (Fig. 4), not readily predicted on the basis of the amino acid inserted or global sequence context

4alone. This suggests the impact of considerable epistatic effects, which were not further considered here. Importantly, the authors place these findings in the context of an existing "redox tuning" model, which clearly shows that all of these adaptations occur without changes in the first and second coordination shells of the metal cofactor, as found in the ground state structure.

Further, all Mn-specific enzymes are resistant to peroxide poisoning; this was not further considered in the context of evolutionary selection.

In general, the text follows logically and the figures are quite good, thus enhancing the impact of the work, which is written appears and clearly accessible to a wide swath of readers across disciplines.

*****END*****

Author Rebuttal to Initial comments

Reviewers' comments:

Reviewer #1 (Remarks to the Author):

In this paper, the authors use a combination of bioinformatics and biochemistry methods to show how the metal preference of Fe/Mn SOD can be finely tuned, and how this actually happened multiple times across evolution. The methods are appropriate, the results are convincing and discussed clearly. Therefore, I recommend publication in Nature Ecology & Evolution.

We would like to thank Reviewer #1 for their review and for their positive comments on the clarity of the text.

Reviewer #2 (Remarks to the Author):

Sendra and colleagues present a very interesting study on the evolution of metal specificity in superoxide dismutases. They employ phylogenetics and extensive biochemistry to show that metal preferences frequently switch and provide some mechanistic insights about how these switches could

5happen. This is one of very few studies that investigate a biochemical phenotype across an entire phylogeny – an approach that I hope will become much more common in the future. The conclusions are interesting and well supported by the data. I believe this work will be broadly interesting to evolutionary biologists and biochemists and as such is highly suitable for the broad readership of Nature Ecology and Evolution.

We share the reviewer's hope that this type of work will become more common in the future. We are grateful for their positive comments as well as the constructive suggestions they have provided.

The manuscript contains extensive amounts of data and is difficult to read at times. One problem is that the paper frequently jumps between different Extended Data figures and that it does not go through them in order.

We have endeavoured to make the text accessible while explaining the extensive data that we present in the manuscript within the constraints of the journal's formatting guidelines. All of the Extended Data were cited in order at first reference, but many of the figures are referred to in multiple places within the manuscript, which reflects the synthetic approach of the study and the richness of the datasets provided in each figure.

A less important problem is that there was no merged document that contained all figures (including Extended Data) with their corresponding legends beneath them. This reviewer found the switching between documents and scrolling very exhausting while reviewing the work.

We understand the reviewer's frustration with the formatting of the manuscript files. This formatting was performed by the journal's submission system, appending the figures to the end of the text file and providing each Extended Data figure as a separate file. To make this version easier to assess, we have modified this to ensure a pdf file is provided containing all figures of the manuscript and their corresponding legends in a single file.

I have a few minor comments:

1) What was the model selection criterion in IQ-tree. I assume it was BIC. This should be described

Yes, the BIC criterion was used for model selection. The selected model, WAG+R10, was the best scoring model under both AIC and BIC criteria. In addition, topologies of the SodFM protein trees generated under the lower scoring LG+G4 model were also consistent with those in the WAG+R10 trees. The details of the model selection criteria have now been added to the methods section entitled “Bioinformatic protein family identification and characterisation”.

2) The trees in the main text figures 1 and 2 lack a scale bar.

Scale bars have now been added to both figures.

3) I would urge the authors to switch from a semi circle representation of the trees to a normal ‘left to right’ representation. Semi circles are incredibly hard to read. I see no advantage of this representation and many disadvantages (including having to tilt my head to read some of the labels and struggling to grasp the relationships between major groups on the tree).

The semi-circular and circular tree representations were used to enable clearer annotation of the associated metadata, which in these large datasets can become unreadable when mapped onto a cladogram format.

In this manuscript, the semi-circular format was chosen when the main focus of the figure was on the metadata that was mapped onto the tree, and where tree topology served mainly as a guide for interpreting said data (Fig1, Fig2, FigExt1, and FigExt4). However, the cladogram formatting was used whenever the tree topology itself was important for the interpretation of the presented data (in Fig3, FigExt5, and FigExt6).

All of the tree files are provided as supplementary files, and can therefore be easily displayed in other formats by readers using tree visualisation software.

74) The cambialism score should be a log score. The authors already plot it on log scales in various figures. It is much easier to compare relative preferences between Fe and Mn on a log scale

We agree that the log scale enables clearer graphical representation of the differences in aCR/CR values, which is exactly why we decided to use this scale in the aCR plots.

However, we do not think that the log score is a clearer or more practical way of representing the cambialism ratio/score in the text. We intuitively use CR values in our daily communications within and between our research groups, and they enable us to intuitively grasp the relative activity with both metals without any confusion. We do not see any added benefit of additional logarithmic transformation because log-transformed values can be more difficult to interpret. For example, $aCR = 2$ clearly indicates that the Fe-dependent activity is 2-fold higher than that with Mn, which is not so obvious when represented as $\log(aCR) = 0.3$. The relative activities with Fe and Mn in aCR measurements rarely differed by more than a factor of 100, therefore the values themselves do not justify the use of log transformation. Log score would also have an additional disadvantage of presenting cambialistic enzymes with value of $\log(CR) = 0$, which would be rather unfortunate for the representation of equal relative activities with two metals and which we find is more intuitively understood when represented as $CR = 1$.

5) Figure 2C overloaded me with different colors and I had to stare at it for a long time to understand what it is supposed to convey. Perhaps this could be simplified.

While we appreciate this is a colourful figure panel, the colour-coding is for an important purpose. The different colours are associated with each of the different amino-acids in Figure 2C, shown on the left of that panel. This colour-code is used to enable a simple comparison between those amino acids at the key positions (the water-coordinating residue and residue X_{D-2}) between panels a, b and c of this figure. We have amended the figure legend to explain this more clearly (lines 484-485).

6) The authors quantify differences in activity to make statements about metal preference. Are there perhaps also differences in metal affinity and do they always correlate with differences in activity? There

8is an assumption in the paper that the observed preferences are somehow optimal for each environment. I'm not sure that statement can be made without knowing the environmental abundances of these metals.

The reviewer highlights the important subject of metal selectivity and its biological significance, which is also an active area of research in the field of metallobiology (Waldron et al., 2009, Glauninger et al., 2018, Young et al., 2021, Choi & Tezcan, 2022). Distinct from metal preference, which reflects the specificity of a metalloprotein's *activity* with distinct metal ions, metal selectivity reflects differences in the relative affinities of *binding* of distinct metals by proteins. Measuring the metal affinities of SodFMs is not feasible because these proteins kinetically trap their metal ion, eliminating metal dissociation. This prevents measurement of the dissociation constant, K_d , which is an equilibrium parameter. Furthermore, such affinity measurements would have to be performed *in vitro* under non-physiological conditions. Physiological metal availabilities inside living cells and how these vary between organisms are largely unknown (Osman et al., 2019). For these reasons, assessing differences in metal selectivity of the SodFMs was outside the scope of this study.

Nonetheless, our data do provide some indication of variation in metal selectivity of SodFMs. Based on our metal-verified preparations (now >100, including those contributing to other forthcoming publications), SodFM metal acquisition can differ when expressed inside the common cytosol of *E. coli* (shown in FigExt7f). Some were fully metal-loaded (e.g. *S. aureus* SodM with both metals) while others do not bind metal (e.g. *E. coli* SodA with Fe). Nonetheless, most of the tested SodFMs' metalation levels were similar, and usually above 50% metal-loaded for both metals (average loading for 30 SodFMs: Mn 78% \pm 26%, Fe 61% \pm 29%). Where metal-loading was lower, it was usually proportionally lower for both metals (average loading in Fe-loaded protein preps relative to Mn-loaded preps was 0.8 \pm 0.2). Furthermore, in multiple X_{b-2} and H/Q mutants we found metal loading was unchanged even where metal preference was altered. Thus, although we agree that metal-selectivity may play important physiological roles, we observed no evidence for it affecting our metal-preference estimations or trends in the roles played by the identified residues.

The hypothesis that metal-preference can be 'optimal' is based on the only experimental system of *S. aureus* where relevant data exists and is consistent with the acquisition of the cambialistic SodFM1 (SodM, aCR=1) being a specific adaptation to surviving metal-starvation (Garcia et al., 2017, Barwinska-

Sendra et al., 2020). To our knowledge, no physiological data for any other system yet exists to further test this hypothesis.

7) The authors make various statements about the evolution of different metal specificities and ancestral specificities, but never use their trees to explicitly reconstruct the evolution of this trait. I think the authors should consider performing a trait reconstruction using their data (i.e. use parsimony or a gaussian model to infer the aCR ratio at internal nodes of their protein tree). The paper would live without it, but since this is a ‘first of its kind’ study, it would be great to set a good precedent.

We appreciate the reviewer’s positivity about our ‘first of its kind’ study. We agree that ancestral reconstruction could be beneficial by providing a model-based test of our hypotheses, for example that the last common ancestors of the SodFM1 and SodFM2 subfamilies likely displayed Mn- and Fe-preferring sequence characteristics, respectively.

To do so, we performed ancestral sequence reconstruction to infer the most likely residues at the identified key positions (X_{D-2} and H/Q_{C-term} or Q_{N-term}) for the ancestral nodes of SodFM1, SodFM2, SodFM3, SodFM4, and SodFM5 last common ancestors, and ancestral aCR state estimation using a sample of our experimentally verified extant SodFMs focusing on the two key nodes of the SodFM1 and SodFM2 last common ancestors. The results of these analyses were consistent with the hypothesis we originally formulated based on the analysis of the metal-preference and residue types across the SodFM tree and distribution of SodFMs across the tree of life.

The results of these additional analyses are now included in the manuscript in Fig2a, where we have added these most likely residues at positions X_{D-2} and H/Q_{C-term} or Q_{N-term} to the ancestral nodes of SodFM1-5, and in Extended Data Fig. 2f and g. We have also added text to the Results section entitled “Metal-preference is not phylogenetically constrained in SodFMs” (lines 94-96) to include the results from these additional analyses. We thank the reviewer for making this suggestion, which we feel has improved the analysis and the manuscript overall.

8) The paper mentions a correlation between different amino acids and the aCR ratio. This language is somewhat unfortunate. To make this statement, a phylogenetic regression would have to be performed.

10A straight correlation ignores the relatedness of the sequences of the tree and can lead to erroneous inferences of correlations. If the authors do not want to perform a phylogenetic regression, they should modify this language.

We agree with the reviewer and have modified the language accordingly (line 141, 149, 155 and 252).

9) The paper uses the term 'evolutionary hotspot' in a manner that I found confusing. I think the authors mean taxonomic groups in which preferences frequently switch. But this term has previously been used to describe regions of proteins where important changes tend to happen. In order to avoid confusion, perhaps the use of this term could be reconsidered.

We agree with the reviewer and have modified the language accordingly (lines 173-176).

10) In line 278 there is a sentence about selection for cambialism. I could not understand how the data support this statement. Either it needs clarification or be removed.

We apologise for not making this unambiguously clear in the previous version of the text. Our data in this section demonstrate that mutation of the *S. aureus* cambialistic SodM can easily yield an Fe-preferring SOD with higher Fe-activity through just a single mutation at the X_{D-2} position. Yet the cambialistic enzyme, and not this Fe-preferring variant, is present in all genomes of *S. aureus* and of its close relative *S. argenteus*. This strongly suggests that the cambialistic isozyme has been selected for, and not merely for an enzyme with Fe-activity. We have more explicitly expanded on this point in the text to make it clearer for the reader (lines 281-286).

11) I found the last few sentences of the discussion a bit over the top. I don't understand how these proteins have anything to do with global warming or epidemics. The work presented here is a major advance in the field of evolutionary biochemistry. I don't think these grandiose statements are necessary to sell it.

We appreciate the reviewer's concern and have revised the final text to focus on the key discoveries of our study (lines 386-399).

Reviewer #3 (Remarks to the Author):

In this manuscript, Dendra et al. present a comprehensive evolutionary analysis of metal preference of the SodFM superfamily of superoxide dismutases that span that entire tree of life. This work derives from previous work by the same investigator team (refs 4-5) and their discovery and characterization of a truly cambialistic SodM from *Staphylococcus aureus*, which is capable of using Fe or Mn as the metal cofactor for superoxide anion dismutation. That prior work lead to the question posed here: What are the evolutionary mechanisms and timescales by which Fe- vs Mn- vs. cambialistic SODs evolved? This is a broadly important question in metalloenzyme catalysis and evolution and the SodFM superfamily is one of a handful of outstanding systems with which to explore this at level of detail required to obtain these insights.

The conclusions that the authors reach appear to be well-supported by the data, and of high interest to those interested in metalloproteomes and metalloenzymology alike. The overall approach is to construct a suitable tree and then purify many (≥ 65) enzymes from each of five main subfamilies (SodFM1-5), or within a single group, and measure the aCR, or approximate cambialism ratio, of recombinant SODs purified as Mn or Fe-laded enzymes.

A couple of key takeaways are well-supported by the data. Phylogenetic grouping on the basis global sequence analysis cannot be used to assess metal-preference. Further, although earliest common ancestors identified here may have been cambialistic or characterized by a particular metal preference, shifts in metal preferences typically occur multiple times during the course of evolution within a main SodFM family. The authors describe this a "sliding scale" of adaptability rather than clearly defined evolutionary path, an important finding.

The authors go on to identify candidate structural features that impact metal specificity, focusing on a few key determinants (XD-2; HCterm/QCterm/QNterm). They provide compelling support for the idea that these residue play important (albeit, not exclusive) roles in metal-preference switching. The importance of the identity of X-2 residue, in particular, was tested biochemically and the model validated. Changes here can push metal preference in one direction exclusively or another or both (Fig. 4), not readily predicted on the basis of the amino acid inserted or global sequence context alone. This

12suggests the impact of considerable epistatic effects, which were not further considered here. Importantly, the authors place these findings in the context of an existing “redox tuning” model, which clearly shows that all of these adaptations occur without changes in the first and second coordination shells of the metal cofactor, as found in the ground state structure.

Further, all Mn-specific enzymes are resistant to peroxide poisoning; this was not further considered in the context of evolutionary selection.

In general, the text follows logically and the figures are quite good, thus enhancing the impact of the work, which is written appears and clearly accessible to a wide swath of readers across disciplines.

We would like to thank Reviewer #3 for their positive review and for recognising the importance of our findings, the accessibility of the manuscript, and its potential impact.

Other changes:

We also made some minor amendments to the figures and text where we spotted typographical errors in the prior versions. We have also added the current affiliation of the last author, who has moved institutions since the original submission, and the new funding that has supported him during the revision period.

Decision Letter, first revision:

13th January 2023

Dear Kevin,

Thank you for submitting your revised manuscript "An ancient metalloenzyme evolves through metal preference modulation" (NATECOLEVOL-221017790A). It has now been seen again by Reviewer #2 and their comments are below. The reviewers find that the paper has improved in revision, and therefore we'll be happy in principle to publish it in Nature Ecology & Evolution, pending minor revisions to comply with our editorial and formatting guidelines.

13Thank you again for your interest in Nature Ecology & Evolution. Please do not hesitate to contact me if you have any questions.

[REDACTED]

Reviewer #2 (Remarks to the Author):

The authors have addressed my concerns and I recommend publication. I appreciate the effort that went into doing both ancestral trait and sequence reconstructions, which were carried out appropriately. I'd like to thank the authors for their patience and congratulate them to great piece of work.

Our ref: NATECOLEVOL-221017790A

16th January 2023

Dear Dr. Waldron,

Thank you for your patience as we've prepared the guidelines for final submission of your Nature Ecology & Evolution manuscript, "An ancient metalloenzyme evolves through metal preference modulation" (NATECOLEVOL-221017790A). Please carefully follow the step-by-step instructions provided in the attached file, and add a response in each row of the table to indicate the changes that you have made. Please also check and comment on any additional marked-up edits we have proposed within the text. Ensuring that each point is addressed will help to ensure that your revised manuscript can be swiftly handed over to our production team.

****We would like to start working on your revised paper, with all of the requested files and forms, as soon as possible (preferably within two weeks). Please get in contact with us immediately if you anticipate it taking more than two weeks to submit these revised files.****

If you have not done so already, please alert us to any related manuscripts from your group that are under consideration or in press at other journals, or are being written up for submission to other

14journals (see: <https://www.nature.com/nature-research/editorial-policies/plagiarism#policy-on-duplicate-publication> for details).

In recognition of the time and expertise our reviewers provide to Nature Ecology & Evolution's editorial process, we would like to formally acknowledge their contribution to the external peer review of your manuscript entitled "An ancient metalloenzyme evolves through metal preference modulation". For those reviewers who give their assent, we will be publishing their names alongside the published article.

Nature Ecology & Evolution offers a Transparent Peer Review option for new original research manuscripts submitted after December 1st, 2019. As part of this initiative, we encourage our authors to support increased transparency into the peer review process by agreeing to have the reviewer comments, author rebuttal letters, and editorial decision letters published as a Supplementary item. When you submit your final files please clearly state in your cover letter whether or not you would like to participate in this initiative. Please note that failure to state your preference will result in delays in accepting your manuscript for publication.

Cover suggestions

As you prepare your final files we encourage you to consider whether you have any images or illustrations that may be appropriate for use on the cover of Nature Ecology & Evolution.

Nature Ecology & Evolution has now transitioned to a unified Rights Collection system which will allow our Author Services team to quickly and easily collect the rights and permissions required to publish your work. Approximately 10 days after your paper is formally accepted, you will receive an email in providing you with a link to complete the grant of rights. If your paper is eligible for Open Access, our Author Services team will also be in touch regarding any additional information that may be required to arrange payment for your article.

Please note that *Nature Ecology & Evolution* is a Transformative Journal (TJ). Authors may publish their research with us through the traditional subscription access route or make their paper

15immediately open access through payment of an article-processing charge (APC). Authors will not be required to make a final decision about access to their article until it has been accepted. [Find out more about Transformative Journals](https://www.springernature.com/gp/open-research/transformative-journals)

Authors may need to take specific actions to achieve [compliance with funder and institutional open access mandates](https://www.springernature.com/gp/open-research/funding/policy-compliance-faqs). If your research is supported by a funder that requires immediate open access (e.g. according to [Plan S principles](https://www.springernature.com/gp/open-research/plan-s-compliance)) then you should select the gold OA route, and we will direct you to the compliant route where possible. For authors selecting the subscription publication route, the journal's standard licensing terms will need to be accepted, including [self-archiving and license to publish](https://www.nature.com/nature-portfolio/editorial-policies/self-archiving-and-license-to-publish). Those licensing terms will supersede any other terms that the author or any third party may assert apply to any version of the manuscript.

[REDACTED]

[REDACTED]

Reviewer #2:

Remarks to the Author:

The authors have addressed my concerns and I recommend publication. I appreciate the effort that went into doing both ancestral trait and sequence reconstructions, which were carried out appropriately. I'd like to thank the authors for their patience and congratulate them to great piece of work.

Final Decision Letter:

15th February 2023

Dear Kevin,

We are pleased to inform you that your Article entitled "An ancient metalloenzyme evolves through metal preference modulation", has now been accepted for publication in Nature Ecology & Evolution.

Over the next few weeks, your paper will be copyedited to ensure that it conforms to Nature Ecology and Evolution style. Once your paper is typeset, you will receive an email with a link to choose the appropriate publishing options for your paper and our Author Services team will be in touch regarding any additional information that may be required

You will not receive your proofs until the publishing agreement has been received through our system

Due to the importance of these deadlines, we ask you please us know now whether you will be difficult to contact over the next month. If this is the case, we ask you provide us with the contact information (email, phone and fax) of someone who will be able to check the proofs on your behalf, and who will be available to address any last-minute problems . Once your paper has been scheduled for online publication, the Nature press office will be in touch to confirm the details.

Acceptance of your manuscript is conditional on all authors' agreement with our publication policies (see www.nature.com/authors/policies/index.html). In particular your manuscript must not be published elsewhere and there must be no announcement of the work to any media outlet until the publication date (the day on which it is uploaded onto our web site).

Please note that *Nature Ecology & Evolution* is a Transformative Journal (TJ). Authors may publish their research with us through the traditional subscription access route or make their paper immediately open access through payment of an article-processing charge (APC). Authors will not be required to make a final decision about access to their article until it has been accepted. [Find out more about Transformative Journals](https://www.springernature.com/gp/open-research/transformative-journals)

Authors may need to take specific actions to achieve [compliance with funder and institutional open access mandates](https://www.springernature.com/gp/open-research/funding/policy-compliance-faqs). If your research is supported by a funder that requires immediate open access (e.g. according to [Plan S principles](https://www.springernature.com/gp/open-research/plan-s-compliance)) then you should select the gold OA route, and we will direct you to the compliant route where

17possible. For authors selecting the subscription publication route, the journal's standard licensing terms will need to be accepted, including <https://www.nature.com/nature-portfolio/editorial-policies/self-archiving-and-license-to-publish>. Those licensing terms will supersede any other terms that the author or any third party may assert apply to any version of the manuscript.

We welcome the submission of potential cover material (including a short caption of around 40 words) related to your manuscript; suggestions should be sent to Nature Ecology & Evolution as electronic files (the image should be 300 dpi at 210 x 297 mm in either TIFF or JPEG format). Please note that such pictures should be selected more for their aesthetic appeal than for their scientific content, and that colour images work better than black and white or grayscale images. Please do not try to design a cover with the Nature Ecology & Evolution logo etc., and please do not submit composites of images related to your work. I am sure you will understand that we cannot make any promise as to whether any of your suggestions might be selected for the cover of the journal.

You can generate the link yourself when you receive your article DOI by entering it here: <http://authors.springernature.com/share>.

[REDACTED]

P.S. Click on the following link if you would like to recommend Nature Ecology & Evolution to your

18librarian <http://www.nature.com/subscriptions/recommend.html#forms>

** Visit the Springer Nature Editorial and Publishing website at http://editorial-jobs.springernature.com?utm_source=ejp_NEcoE_email&utm_medium=ejp_NEcoE_email&utm_campaign=ejp_NEcoE for more information about our career opportunities. If you have any questions please click [here](mailto:editorial.publishing.jobs@springernature.com).**